# Temporal horizons in forecasting: a performance-learnability trade-off

Pau Vilimelis Aceituno[1,*], Jack William Miller[2], Noah Marti[1], Youssef Farag[1], and Victor Boussange[3]

[1]Institute of Neuroinformatics, ETH Zürich and University of Zürich, Winterhurerstrasse 190, Zürich 8057, Switzerland
[2]School of Computing, National Australian University, 108 North Rd, Acton ACT 2601, Australia
[3]Unit of Land Change Science, Swiss Federal Research Institute for Forest, Snow and Landscape Zürcherstrasse 111, Birmensdorf 8903, Switzerland
[*]Corresponding author: Pau Vilimelis Aceituno, pau@ini.uzh.ch

**Reviewed on OpenReview:** `https://openreview.net/forum?id=BeudQIxT1R`

## Abstract

When training autoregressive models to forecast dynamical systems, a critical question arises: how far into the future should the model be trained to predict for optimal performance? In this work, we address this question by analyzing the relationship between the geometry of the loss landscape and the training time horizon. Using dynamical systems theory, we prove that loss minima for long horizons generalize well to short-term forecasts, whereas minima found on short horizons result in worse long-term predictions. However, we also prove that the loss landscape becomes rougher as the training horizon grows, making long-horizon training inherently challenging. We validate our theory through numerical experiments and discuss practical implications for selecting training horizons. Our results provide a principled foundation for hyperparameter optimization in autoregressive forecasting models.

## 1 Introduction

Forecasting the future state of dynamical systems is a fundamental challenge in scientific and engineering disciplines, from climate modeling to robotics. A central objective is to generate accurate predictions as far into the future as possible. Autoregressive (AR) models, which iteratively feed their own predictions back as inputs, are a standard tool for this task(Petropoulos et al., 2022; Lai & Lu, 2017; Azencot et al., 2020; Van den Oord et al., 2016; Lam et al., 2023).

Traditional AR models were linear and limited to low-dimensional time series (Petropoulos et al., 2022; Lai & Lu, 2017). Modern AR models based on neural networks and trained via gradient descent, have emerged as powerful alternatives due to their ability to capture nonlinear dynamics in high-dimensional systems. Such nonlinearity is ubiquitous across scientific domains, including climate science (Houghton et al., 2001), ecology (Benincà et al., 2009; Huisman & Weissing, 1999), physics (Levien & Tan, 1993), and robotics (Iqbal et al., 2014). Consequently, AR models have demonstrated success in e.g. weather and climate forecasting (Lam et al., 2023; Kochkov et al., 2024), ecological forecast (Boussange et al., 2024), physical system forecast (Azencot et al., 2020; Erichson et al., 2019; Li et al., 2020; Miller et al., 2022), or robotic control (Sasagawa et al., 2021; Williams et al., 2017; Heetmeyer et al., 2023). Beyond forecasting, AR models can be used for generative tasks like text or code generation (Brown et al., 2020).

A critical yet understudied challenge in training AR is selecting an appropriate temporal forecast horizon during training. Common approaches either use a single step prediction scheme (Van den Oord et al., 2016; Azencot et al., 2020; Miller et al., 2022; Brown et al., 2020) or adopt longer horizons without rigorous justification (Lam et al., 2023; Kochkov et al., 2024; Price et al., 2025; Cornille et al., 2024). Currently,

no theoretical framework guides this choice. This gap extends to mechanistic modeling with differential equations or discrete maps, where methods like piecewise regression (Pisarenko & Sornette, 2004; Doya et al., 1992; Boussange et al., 2024) and multiple shooting (Bock, 1981; England, 1983; Aydogmus & Tor, 2021), which similarly lack principled criteria for horizon selection.

In this work, we rigorously analyze how the temporal forecast horizon during training affects AR model performance in the context of dynamical system prediction. We derive practical guidelines for selecting this horizon based on the underlying system dynamics. Our contributions include:

- a theoretical analysis linking the temporal forecast horizon during training to the geometry of the loss landscape,

- an empirical validation of these relationships across different dynamical systems,

- practical recommendations for horizon selection in both data-driven and mechanistic modeling.

The remainder of this paper is organized as follows. We first situate our work within the existing literature. Next, we introduce necessary background and notation. We then present our theoretical analysis of how temporal horizons shape the loss landscape, that we illustrate with numerical experiments. We examine the practical implications of our theoretical findings in the presence of noise, and how it connects with the task of fitting mechanistic models. Finally, we conclude and suggest future research directions.

## 2 Related works

Most of the theoretical literature on machine learning for dynamical system forecasting has focused on the vanishing or exploding gradient problem (VEGP), a phenomenon where the gradients over the parameters of a neural network scale with the temporal distance between the current and past state of the network (Pascanu, 2013; Hochreiter et al., 2001; Bengio et al., 1994). Common strategies that have been proposed to mitigate VEGP include architectural modifications (e.g., gated mechanisms (Hochreiter, 1997)), specialized weight initializations (Narkhede et al., 2022), gradient clipping (Goodfellow, 2016; Zhang et al., 2020), and constrained dynamics (e.g., unitary recurrent neural networks (Arjovsky et al., 2016; Chang et al., 2019; Erichson et al., 2020)). However, these techniques limit the expressivity of the models or introduce biases (Orhan & Pitkow, 2019; Schmidt et al., 2021; Mikhaeil et al., 2022). Recent works have derived a connection between the VEGP and system properties like Lyapunov exponents (Mikhaeil et al., 2022), which provides an intuition of how fast gradients explode, suggesting that it is possible to analyze mathematically the connection between machine learning for forecasting and the dynamics of the system being learned.

On the practical side, tuning the forecast horizon is an empirical technique that has been used in a variety of context to optimize the performance of AR models. In particular, this strategy is also referred to as multiple shooting or piecewise inference, and has been proposed for fitting classical discrete maps (Pisarenko & Sornette, 2004), differential equation models Boussange et al. (2024), transformer models applied to language (Gloeckle et al., 2024; Cornille et al., 2024) and in the context of reinforcement learning Zhang & Sutton (2017). Similar strategies implicitly tune the forecast horizon via discount factors, effectively reducing the relevance of long time horizons. Concrete examples can be found for example in weather forecasting (Lam et al., 2023), financial time series modeling (Bernaciak & Griffin, 2024), or in reinforcement learning, where the effective horizon influences model success, for example through discount factors in techniques like $\lambda$-returns (Sutton, 1995; Anand & Precup, 2021). The relevance of temporal horizons for forecasting also appears in other fields. Model predictive control also requires careful horizon selection (Hewing et al., 2020). Similarly, neural PDE solvers often prioritize high-frequency components, which are inherently tied to shorter temporal dependencies (Wiener, 1930; Lippe et al., 2023; Kurth et al., 2023).

Thus, although it is well documented that the training horizon has an effect on the convergence of AR models, we do not know of theoretical work that directly investigates the effect of the forecast horizon on the geometry of the loss landscape.

# 3 Background and Notation

## 3.1 Dynamical systems

Dynamical systems consist of a state space $\mathcal{X}$ in which the system has one position $x$, and a mathematical function that describes how $x$ changes with respect to $t$. As the state of the system evolves in time, it defines a trajectory. Trajectories may show a variety of patterns. For example, they can converge to an attractor, diverge from a point of manifold, remain within a manifold indefinitely. Some trajectories are inherently easy to predict (e.g., trajectories converging to a fixed point); some cannot be predicted (e.g., unstable systems, which are constantly expanding into new states).

Two particularly interesting families of trajectories, limit cycles and chaotic attractors, appear exclusively in nonlinear systems and are neither trivially predictable nor impossible to predict, and will be the focus of this work.

- **Limit cycles** are orbits that repeat themselves, and can be characterized by the a closed line that defines their orbit and by their period $p$ or their **frequency** $\omega = \frac{2\pi}{p}$, which represents the time it takes for them to come back to a previous point; such that $x_{t+p} = x_t$ (Brin & Stuck, 2002).

- **Chaotic trajectories** on the other hand never repeat any point in the trajectory, and can be defined by their chaotic attractor (the set of points towards which the system will evolve). In chaotic trajectories, the **Lyapunov exponent** $\lambda$ measures how two points that are initially close in the attractor diverge with time by the relationship $|x_t - x_t'| \approx e^{\lambda t}|x_0 - x_t'|$, where $x_0$ is infinitesimally close to $x_0'$(Brin & Stuck, 2002).

Crucially, both chaotic or limit cycle systems are practically impossible to forecast very far into the future, as small differences between the model and the system being modeled will make long term predictions essentially random (see A.5).

## 3.2 Examples of dynamical systems

In order to develop and illustrate our theory, we will use the following examples of dynamical systems: A Lorenz attractor, a double pendulum, a simple limit cycle, and a chaotic food web model Hastings (1991). The Lorenz attractor, food web and the Double pendulum are classical examples of chaotic attractors. All the systems are described in detail in appendix C.

## 3.3 Autoregressive models

Trajectories in discrete time dynamical systems, or continuous dynamical systems whose trajectories are sampled at fixed intervals, can generally be characterized by a function that expresses how the system evolves in a single timestep,

$$x(m) = \phi(x(m-1)) \tag{1}$$

where $m$ is the temporal index. Under this formulation, the dynamics of the system can be captured by a function $f$, such that for a given state $x(m)$, $f(x(m), \theta)$ gives a subsequent state $x(m+1)$ or an approximation thereof.

Since $f$ is constructed to output vectors with the same dimensionality as the input, we can generate a future state $x(m+k)$ by simple composition of the model

$$x(m+k) = f \circ \cdots \circ f(x(m), \theta) = f^k(x(m), \theta). \tag{2}$$

where $k$ corresponds to the number of compositions. This composition constitutes the simplest definition of AR (to use a model's own predictions to generate further predictions), and we will thus refer to the model implementing $f$ as autoregressive. Note we can extend the output vectors to include hidden states or context windows (see A.20), thus covering RNNs and Transformers. However, doing so would require significantly more mathematical machinery we leave it for future work (see remark A.21).

To simplify our theory and experiments we will focus on models where the function $f$ is implemented by a feedforward neural network which we will refer to as autoregressive neural network (ARNN), and implement it with multi-layer perceptrons for our experiments (MLP, see appendix F for details about the models).

### 3.4 Loss functions for ARNNs

Consider a time series of discrete states $\mathbf{x} = [x(0), x(1), \ldots, x(M)]$ sampled at times $[0, 1, 2, ..., M]$. We can define a quadratic ARNNs loss as

$$\mathcal{L}_{\mathbf{x}}(\theta, T) = \frac{1}{M-K} \sum_{m=1}^{M-K} \frac{1}{K} \sum_{k=1}^{K} \left\| x(m+k) - f^k\left(x(m), \theta\right) \right\|^2. \tag{3}$$

where $K$ indexes the number of autoregressive steps ($k$ in eq. (2)) required to forecast the state of the system in a temporal horizon $T$ which uses the temporal units of the dynamical system in question. Note that the temporal units for $T$ depend on the underlying system being studied and can be continuous, while the autoregressive steps $K$ depends on the choice of sampling rate and are natural numbers, and we will usually refer to $T$ in our paper. We will use the Euclidean norm squared to derive bounds on errors and losses for the rest of the paper, but other norms could also be used. Note that it is possible to have different temporal horizons ($T$s) for training and testing. Unless explicitly stated, we will focus our study on the temporal horizon during training (not testing).

### 3.5 Theoretical assumptions and model set-up

To make our derivations we make three main assumptions:

- We assume that the loss function is smooth, which implies differentiability and we can therefore use gradient descent.

- We assume that we have enough data during training. In terms of dynamical systems, this means that we have sample trajectories that are long enough to evaluate various temporal horizons, and that the data provide a comprehensive coverage of the stationary distribution of the dynamical system.

- We assume that our dynamical systems are ergodic, meaning that a long enough trajectory will converge to a stationary distribution in which every point is visited with a fixed probability. We also assume that our systems are fully observable, and deterministic, meaning that every observation corresponds to a unique point in the state space of the system from which it is possible to predict the next.

Those assumptions allow us to derive our main results, but might not be realized in the real world. In section 4.4 we discuss how and when those assumptions can be relaxed and how to extend our results to other architectures such as RNNs and Transformers.

## 4 Temporal horizons and loss landscapes

In this section, we derive a connection between the loss gradient and the forecast horizon during training, and exploit it to understand the geometry of the loss landscape. Full proofs of our theorems and their corollaries are presented in the appendix A; in the main text we only provide an outline. We provide simulations to illustrate all our theorems.

### 4.1 The building blocks to link dynamics and training loss

To build our theory we need to characterize the region on the parameter space where the model $f(\cdot, \theta)$ is a reasonable approximation of the true dynamics $\phi$. We do so by limiting how far the model can deviate from the true dynamics, which we characterize by the Jacobian matrix – the matrix of partial derivatives of the system that captures how the system evolves locally.

**Definition 4.1** ($\epsilon$-bounded region). For $\epsilon > 0$, an $\epsilon$-bounded region of the parameter space $\Theta_\epsilon$ for a model trained to forecast a dynamical system with a bounded stationary distribution is a convex set within the space of parameters of the model such that

$$\|f(x + \epsilon\vec{r}, \theta) - (f(x, \theta) + J_\phi(x)\vec{r}\epsilon)\| < \epsilon^2 \quad \forall x \in \mathbf{S_x}$$

where $J_\phi$ is the Jacobian matrix of $\phi(\cdot)$, $\vec{r}$ is a unitary random vector and $\mathbf{S_x}$ is the region of the state space where the stationary distribution has non-zero probability.

*Remark* 4.2. Notice that the definition of $\epsilon$-bounded region is closely related to the notion of a ball around a point as it is often used in a geometric treatment of dynamical systems Jost (2005). However, the metric is defined through the error of the model, which is, to the best of our knowledge, a new approach. This connection will allow us to leverage the theory of dynamical systems to infer properties of the loss function in the rest of the paper.

We note that $\epsilon$-bounded regions exist for arbitrarily small $\epsilon$ (lemma A.1), and have bounded loss (lemma A.7). Importantly, a small $\epsilon$ can only be achieved in models that have been partially trained, so that they are already a good approximation of the system dynamics. Thus, our theory applies to models that have been at least partially trained, not to models at initialization.

Now we can connect the dynamics of the system to the loss of a model trained to forecast the system. Consider the function

$$g(T) = \frac{\|\nabla_\theta \mathcal{L}_\mathbf{x}(\theta, T)\|}{\|\nabla_\theta \mathcal{L}_\mathbf{x}(\theta, 1)\|} \tag{4}$$

which represents the relative scaling of the gradient with respect to $T$. The following proposition and its validation in fig. 1 shows how $g$ evolves with $T$. Note that we used 1 as the reference for the gradient magnitude, but any finite value would work.

**Proposition 4.3** (Loss gradient growth). *Consider a dynamical system which is either chaotic, limit cycles, and a corresponding model with an $\epsilon$-bounded region of the parameter space $\Theta_\epsilon - \{\Theta_{\min}^\rho\}$ where $\Theta_{\min}^\rho$ are all the balls of radius $\rho \ll \epsilon$ around the minima of the loss. For a sufficiently small $\epsilon$, when the forecasting horizon $T$ is large, $g(T)$ scales with $T$ as*

$$g(T) = \begin{cases} \mathcal{O}(e^{\lambda T}), & \text{for chaotic/unstable} \\ \mathcal{O}(\omega T), & \text{for limit cycles.} \end{cases} \tag{5}$$

*Proof sketch.* The Jacobian of the system over a time window $T$ will scale exponentially or linearly with $\lambda$ or $\omega$. Furthermore, within definition 4.1, the Jacobian of the model remains close to the Jacobian of the true system (lemma A.6). The Jacobian of the model is always contained in the expression for $\nabla_\theta \mathcal{L}_\mathbf{x}(\theta, T)$, and thus it scales with $T$. See A.8 for the full proof. Note that we removed the minima in $\{\Theta_{\min}^\rho\}$ to avoid zeros, which would lead to a divergence of $g$. We will treat them in the next corollary. □

An important limitation of proposition 4.3 is that it does not inform us about the minima of the loss. Yet, we can use it to gain insights into the Hessian of the loss around those same minima.

**Corollary 4.4.** *Consider a dynamical system which is either chaotic or a limit cycle, and a corresponding model with with minima in an $\epsilon$-bounded region of the parameter space $\Theta_\epsilon$. For a sufficiently small $\epsilon$, when the forecasting horizon $T$ is large, the Hessian around a minima scales with $T$ as*

$$\frac{\|H(\theta, T)\|^*}{\|H(\theta, 1)\|^*} = \begin{cases} \mathcal{O}(e^{\lambda T}), & \text{for chaotic systems} \\ \mathcal{O}(\omega T), & \text{for limit cycles} \end{cases} \tag{6}$$

*where $\|H(\theta, \cdot)\|^*$ is the sum of the eigenvalues of the Hessian (which in a minima is equivalent to the so-called Nuclear norm).*

*Proof sketch.* As the gradient grows with $T$, the second derivative must necessarily grow. To make this formal, we consider small balls around the minima, and use the divergence theorem, where the gradient is the vector field and the trace of the Hessian emerges. □

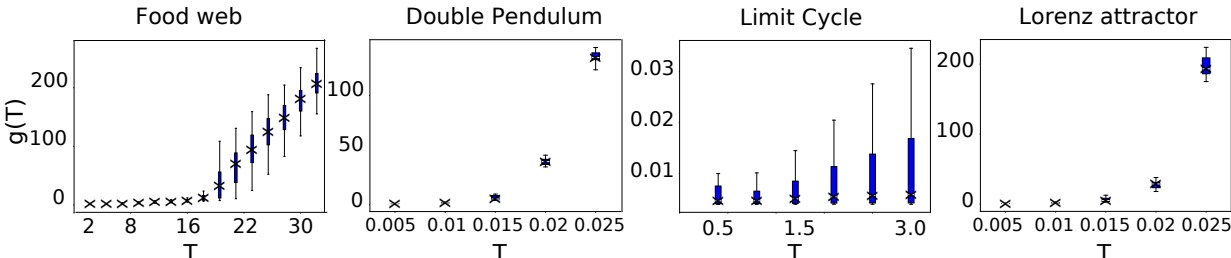

Figure 1: **Gradient scaling**: We measured the relative $L_2$-norm of the gradients ($g(T)$) as a function of $T$ for MLPs $g(T)$ during training. As expected from proposition 4.3, we observe an exponential increase of $g(T)$ for the lorenz system and the double pendulum, and a linear one for the limit cycle and the food web. Notice that we used the temporal units of the system dynamics as units for the temporal horizon to make our results independent of the choice of timestep size, and used a single prediction as the default timestep.

*Remark* 4.5. Note that proposition 4.3 are a direct consequence of dynamical systems theory, and they have appeared previously in the study of VEGP (see for example Mikhaeil et al. (2022), Thm. 2 and 3). However, we link them to the geometry of the loss landscape through the Hessian, which we later will extend by leveraging definition 4.1.

Corollary 4.4 lends itself to two interpretations:

- Classical statistics suggests that having a minimum with a high Fisher information – a quantity that scales with the Hessian – means that the model extracts a lot of information from the data Ly et al. (2017). This would suggest that it is better to pick a high $T$ as it would make any minima found very precise.

- Flat minima seem to be better at generalization, because any difference in parameters (for example due to differences between training and testing data) will have little effect on the loss Hochreiter & Schmidhuber (1997); Keskar et al. (2016). This would suggest that it is better to pick a low $T$ to improve generalization.

Clearly, both intuitions are at odds with each other. To gain further insights we will study the geometry of the loss landscape.

### 4.2 Generalization across temporal horizons

A key problem in comparing models trained with high and low $T$ is that different forecast horizons correspond to different forecasting problems, and therefore we should not directly compare their minima in terms of their loss or forecasting performance. Instead, we need a metric that directly relates the performance of a model trained to one temporal horizon to its performance in another temporal horizon.

We thus start by asking whether a minima found for a given time horizon will work for another time horizon. We formalize this notion as the ratio between two minima for different time horizons,

$$r(T_h, T_l) = \frac{\mathcal{L}_{\mathbf{x}}(\theta_h^{\min}, T_h) - \mathcal{L}_{\mathbf{x}}(\theta_l^{\min}, T_h)}{\mathcal{L}_{\mathbf{x}}(\theta_l^{\min}, T_l) - \mathcal{L}_{\mathbf{x}}(\theta_h^{\min}, T_l)} \tag{7}$$

where the parameter values $\theta_h^{\min}, \theta_l^{\min}$ are two minima found with losses evaluated at $T_l$ and $T_h$ respectively with $T_l < T_h$ ($l$ stands for low and $h$ for high).

Note that different temporal horizons could have different numbers of minima and those might be clustered in different regions of parameter space, and therefore we need to restrict $r$ to comparable minima. Thus, we will

focus on specific pairs $(\theta_\mathrm{h}^{\min}, \theta_\mathrm{l}^{\min})$, where if the model is initialized with parameters $\theta_\mathrm{h}^{\min}$ and trained with the temporal horizon $T_\mathrm{l}$, then it would converge to $\theta_\mathrm{l}^{\min}$, and conversely, if the model starts with parameters $\theta_\mathrm{l}^{\min}$, it would converge to $\theta_\mathrm{h}^{\min}$, if trained at $T_\mathrm{h}$.

**Theorem 4.6** (Minima with long forecasting horizons generalize to lower horizons)**.** *Consider a model $f(\cdot, \theta)$ with $\theta$ in an $\epsilon$-bounded region. We assume the existence of two minima $\theta_l^{\min}$ and $\theta_h^{\min}$ for the losses $\mathcal{L}_\mathbf{x}(\theta, T_l)$ and $\mathcal{L}_\mathbf{x}(\theta, T_h)$ respectively with $T_h > T_l$ such that $\theta_l^{\min}$ would converge to $\theta_h^{\min}$ if trained on $T_h$, and vice-versa. Then the difference in the change in losses follows*

$$r(T_h, T_l) = \begin{cases} \mathcal{O}(e^{\lambda(T_h - T_l)}), & \text{for chaotic/unstable} \\ \mathcal{O}(\omega(T_h - T_l)), & \text{for limit cycles} \end{cases} \tag{8}$$

*Proof Sketch.* We can follow the gradient of the loss for $T_\mathrm{h}$ or $T_\mathrm{l}$, between the two minima. The change in loss is the integral of that gradient, which scales as proposition 4.3. Taking the ratio of the aforementioned change in loss yields our result. See A.10 for the full proof. □

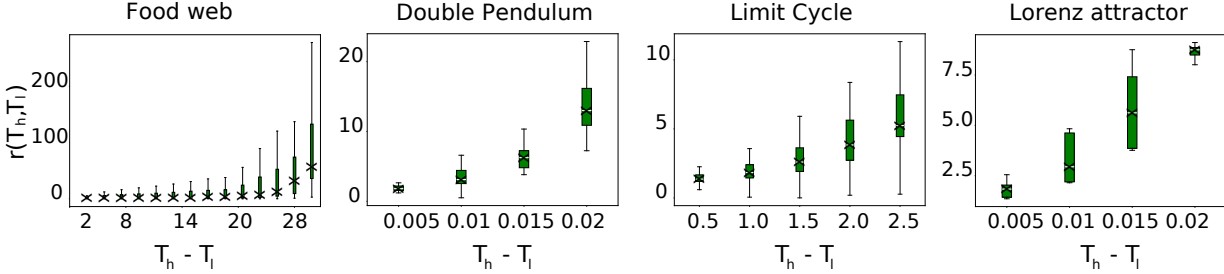

Figure 2: **Performance ratios for longer temporal horizons**: We measured the ratio of the difference between losses on connected minima found at different temporal horizons $r(T_\mathrm{h}, T_\mathrm{l})$, and plotted it against the difference between those temporal horizons. We computed the loss by training a single model for $T_\mathrm{l}$ corresponding to one forecasting step, and then we trained that model further with higher $T_\mathrm{h}$. As expected from theorem 4.6, we observe an upward trend for all the systems, with the limit cycle seemingly linear, the double pendulum and food web being similar to an exponential and the Lorenz attractor being somewhat inconclusive. Notice that here the difference in losses is implicitly linked to the accuracy of the minima we find, which depends on the optimizer and the randomness of the system, thus it is expected that we do not find a perfect match to our theory.

Intuitively, the theorem reflects the fact that long temporal predictions rely on shorter predictions, implying that models that are able to make predictions over long temporal horizons must also make good predictions over short time horizons. However, the converse is not necessarily true, and valid short-term predictions might diverge in the long run, suggesting that $\theta_\mathrm{h}^{\min}$ might better capture the global dynamics of the system than $\theta_\mathrm{l}^{\min}$.

### 4.3 Loss landscape roughness and the temporal horizon

Although theorem 4.6 would suggest that we should train models with a long $T$, we need to beware that such training might be extremely hard. We quantify this through a notion of roughness, which we take as the number of maxima and minima of the loss $z(T, \theta_1, \theta_2)$ found over a line in parameter space $\overrightarrow{\theta_1 \theta_2}$, and prove that it scales with the temporal horizon in theorem 4.7, which is validated empirically in fig. 3.

**Theorem 4.7** (Loss landscape roughness)**.** *For any two points $\theta_1, \theta_2$ in an $\epsilon$-bounded region of the parameter space that are not in a connected region of zero loss[1], the number of minima and maxima along the line*

---

[1]such scenario is not covered in the assumptions on proposition 4.3, and in this case there is no learning

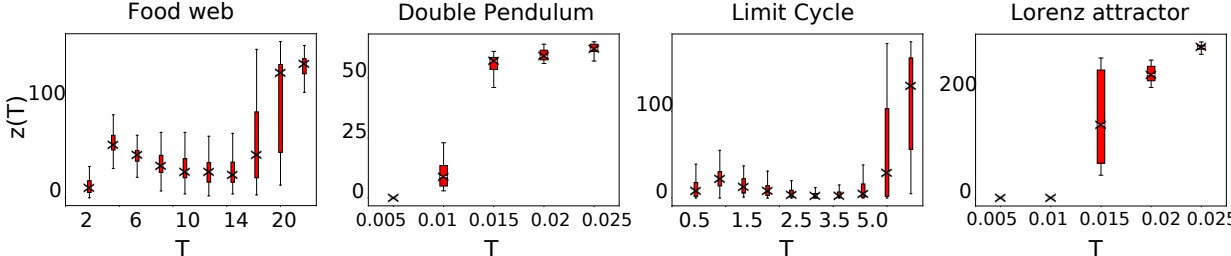

Figure 3: **Loss landscape roughness**: We counted the number of minima and maxima on a cross-section between two parameter sets $\theta_1$, $\theta_2$ achieving low losses, for different $T$ values, denoted as $z(T)$. To find $\theta_1, \theta_2$, we trained a single model with one forecasting step ($k = 1$), and we took late training stages which had low loss but with training epochs in-between to ensure that there is a distance between the two. As expected from theorem 4.7, we see a clear increase in the number of zeros found. Note the number of minima and maxima that we can detect is limited by how finely we can discretize the line between $\theta_1$, and $\theta_2$, and since we have to evaluate the loss at each point in that line, our results underestimate the number of cross-section minima. This explains the saturation seen in the Double Pendulum and the Lorenz attractor which is not expected from our theory.

*segment that connects them will grow as*

$$z(T, \theta_1, \theta_2) = \begin{cases} \mathcal{O}(e^{\lambda T}\|\theta_1 - \theta_2\|), & \text{for chaotic/unstable} \\ \mathcal{O}(\omega T\|\theta_1 - \theta_2\|), & \text{for limit cycles} \end{cases} \tag{9}$$

*Proof Sketch.* Consider two arbitrary elements $\theta_1, \theta_2$ in an $\epsilon$-bounded region. We consider the variation in the loss function through a line segment $l$ connecting $\theta_1$ and $\theta_2$, which takes the form of an integral dependent on $|\partial_l \mathcal{L}_\mathbf{x}(l, T)|$. This derivative is the projection of $\nabla_\theta \mathcal{L}_\mathbf{x}(l, T)$ onto $\vec{u}_{\theta_1 \to \theta_2}$, the unit vector of the line segment, and thus grows with $T$ as shown in proposition 4.3. As $T$ grows, the loss variation grows beyond the maximum possible loss $\mathcal{L}_\epsilon^{\max}$, thus it must go up and down, and hence have a minimum or maximum. By letting $T$ grow further, it will have more and more minima and maxima. See A.12 for the full proof. □

*Remark* 4.8. Notice that the minima and maxima in this theorem are defined as local minima in the line between $\theta_1$ and $\theta_2$, not local minima of the loss function in its full dimensionality. It would possible to have the same number of local minima as $T$ grows. For example, in a valley with a single minima winding around the line segment. The theorem does refer to local minima in single parameter training (see section 5.3).

A curious corollary from this theorem shows that as the temporal horizon grows towards $T \to \infty$, the loss landscape becomes a fractal (corollary A.13). Since a fractal is non-differentiable, this result implies that gradient descent methods would inevitably fail. In the more realistic case of $T < \infty$, theorem 4.7 simply states that the loss landscape becomes harder to navigate as $T$ grows.

## 4.4 Limitations, generalizations and implications

To develop our theory we assumed that forecasting is applied to fully observable systems that are deterministic and where we have data covering all the relevant state space. Such assumptions make our theory concise and amenable to experimental verification, but are often invalid in realistic scenarios. In this section we discuss how and when our assumptions can be relaxed.

First, we acknowledge that some assumptions such as ergodicity, stationarity and broad state coverage are fundamentally linked to statistical learning, and thus cannot be avoided in a mathematical treatment of forecasting (see appendix B).

In partially observable dynamical systems, it might be impossible to make forecasts from single observations (ex: we cannot infer the speed of an object from a single observation of its position). However, multiple delayed observations can contain enough information to recover the full state of the system Takens (1981b;a), as it is often done for Transformers in forecasting tasks (Nie et al., 2022) (see lemma A.16), but also in classical deep neural networks Deco & Schürmann (2012). Another approach is to use hidden states, which can keep information from past observations to recover the state, as done in RNNs (see lemma A.18). Extending our theory to those architectures would thus relax this assumption (see proposition A.20).

A more fundamental problem is that we assume deterministic dynamics, but real data are often stochastic. Yet, stochasticity is not an all-or-nothing phenomena, as there are systems that are very stochastic, and systems that are almost deterministic. Our theory should partially hold for systems with sufficiently small noise, but not for systems that are too noisy to be predicted even with $T = 1$ in the first place. More specifically, our theory remains approximately valid within a time window given by the noise level and the system dynamics (see proposition A.14), after which the system is too random to be forecast. We verify this in the ecology model, where we observe that indeed the noise limits the optimal forecasting horizon (see appendix G).

Finally, we discuss the practical implications of our theory. Our loss surface analysis suggests that increasing $T$ will lead to better minima by theorem 4.6. However, as a result of theorem 4.7, we would expect that a large value of $T$ will render learning impractical. In the next section we explore in more depth this trade-off, how it applies to real data, and other related implications.

## 5    Practical insights

To connect our theory with practical problems, we (1) investigate the effects of the temporal forecasting horizon $T$ on the performance of ARNNs for forecasting the synthetic dynamical systems presented in the previous section and on three empirical datasets, (2) discuss challenges in choosing the training temporal horizon $T$ and the factors that affect it, and (3) place our theory in the context of fitting the parameters of mechanistic models.

### 5.1    The temporal horizon as an hyper-parameter

We trained residual MLPs with different training temporal horizons for the four dynamical systems presented in appendix C until they appeared to reach convergence or a large total wall time cutoff. We evaluated their performance in forecasting using the mean square error (MSE), which corresponds to the testing loss in eq. (3).

Based on our theory, we would expect to observe the following set of behaviors: (1) an initial increase in the performance of models with larger prediction horizons (theorem 4.6) but (2) worse performance for models trained on time horizons sufficiently long such that the loss landscape becomes harder to navigate (theorem 4.7). We find the expected U-shaped curve in all the dynamical systems considered, as seen in fig. 4. Importantly, the optimal training horizon is neither one step nor the time horizon that we wish to use for forecasting.

To evaluate whether our theory applies beyond the original synthetic dynamical systems, we trained similar residual MLPs on three real-world datasets:

- The ClimSIM dataset (see appendix D for details), which is a complex deterministic simulation of weather patterns.

- The National Oceanographic Atmospheric Administration Sea Surface Temperature dataset (NOAA SST, Huang et al. (2021)), which comes from real world measurements and thus contains noise.

- The AMZN (Amazon) stocks from 1997 to 2017 (Kouroupetroglou, 2019), which contains a short dataset that is noisy and non-stationary.

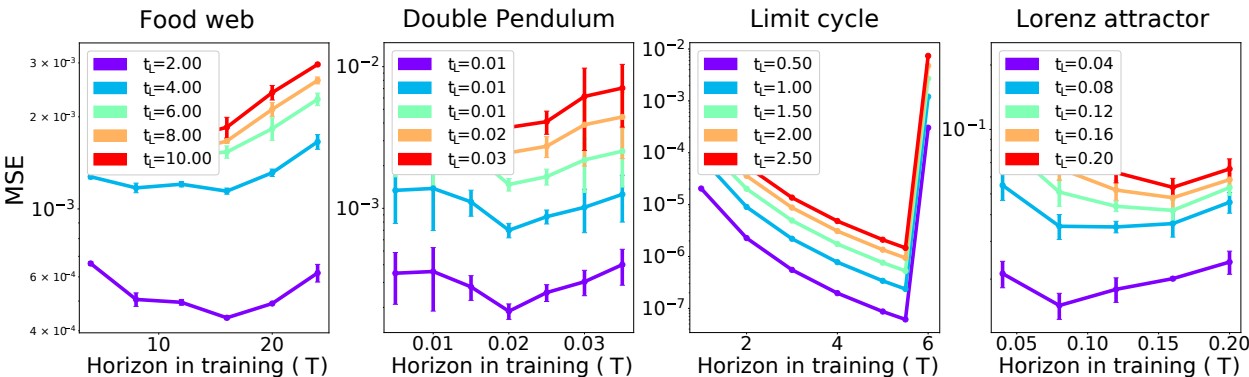

Figure 4: **Performance of residual MLPs trained to predict four dynamical systems with MSE on the $y$-axis and the temporal horizon $T$ used in training on the $x$-axis**. For each model, we took the minimum validation MSE. Each line in the figure is associated with a given evaluation horizon marked by $T_l$. Each point as one goes along the line represents the median performance of a model (with interquartile range error bars) trained with an increasing training horizon $T$ when evaluated $T_l$ time into the future. That is, if we have a model $f(x, \theta_T)$ trained on $T$, the points above $T$ on the graph are $\|x(T_l) - f^{n_t}(x(0), \theta_T)\|$ where $n_t$ is the number of auto-regressive steps to get to $T_l$. As expected, the loss grows with $T_l$, the loss is convex, and the optimal predictive horizon for training rarely coincides with $T_l$.

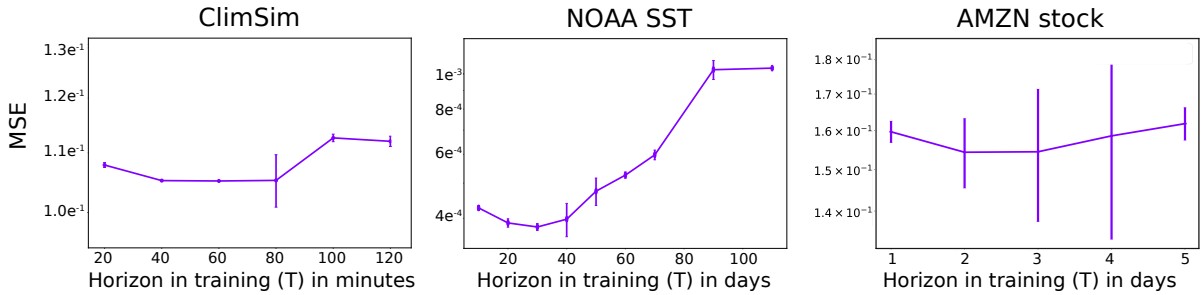

Figure 5: **Performance of various architectures trained with different training horizons on the ClimSim, NOAA SST, and AMZN stocks datasets**. We evaluated the average performance 5 time steps into the future for both tasks, which correspond to 50 days in the case of SST and 100 minutes in the case of ClimSim. We plotted the median of the MSE with upper and lower quartiles. The loss is convex with respect to the temporal horizon in training, and the optimal training horizon is not the evaluation horizon. Note that the AMZN stock shows high forecasting variability, as expected from a noisy, non-stationary time series

Together, the three datasets consist of examples where dynamics fulfill all of our original assumptions but are very complex (the ClimSIM dataset), a dataset that is well behaved (stationary, ergodic, full coverage) but noisy (the NOAA SST dataset), and a dataset where our core assumptions are not valid (the AMZN stock dataset; financial data is considered non-stationary, and the range of stock values goes beyond those in the training data). In all cases we find that the optimum training horizon is longer than a single step-ahead.

Crucially, the optimal training time horizons are not equivalent to the testing horizon, further suggesting that the optimal training horizons are inextricably linked to the system's dynamics rather than the desired prediction horizon. This was found not only on our simple dynamical systems, but also on more complex benchmark datasets on climate modeling (an adapted version of the ClimSim Yu et al. (2023), the sea surface temperature data National Oceanic and Atmospheric Administration Huang et al. (2021), and the Amazon stock prices Kouroupetroglou (2019)).

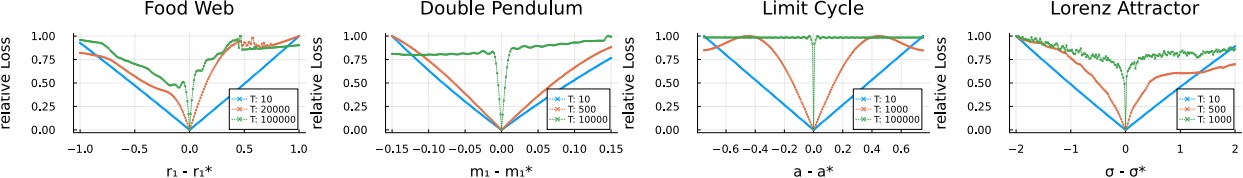

Figure 6: Normalized values of $\mathcal{L}_\mathbf{x}(\theta, T)$ from eq. (3) when altering one dimension of $\theta$ for different values of $T$ in several dynamical systems models. The loss is normalized for each $T$ so that its maximum value is 1. For all the systems, the loss is generally flat with a single minima for low $T$ and becomes steeper and has more minima for large $T$.

As we show the existence of a non-trivial optimal $T$ for both synthetic and realistic datasets, the next natural question is how to find it. We discuss this problem in the next section.

## 5.2   Optimal temporal horizons

Our theory suggests that the upper limit on $T$ (theorem 4.7) is set by the roughness of the loss landscape. While it is possible to find good minima in rough loss landscapes, but doing so requires significant computational resources. Thus, increasing the computational resources should lead to higher optimal temporal horizons. We tested this in appendix H for the ecology system, finding that the computing time used for training does indeed correlate with the optimal temporal horizon for training $T$.

This finding is problematic, because a temporal horizon $T$ is only optimal for a given compute budget, making naive hyperparameter search methods very expensive. For example, grid search methods usually train models with different parameter values and a limited amount of computational resources, and based on the results select the best combination of hyperparameters for a longer training. However, this process becomes very expensive if the hyperparameter depends on the computational resources, because the optimal hyperparameters would only be found if each step in the hyperparameter search uses a similar amount of computational resources as in the final training after the hyperparameter search.

A potential solution would be to use an iterative algorithm that slowly increases $T$ while decreasing the learning rate. To test this, we designed a simple scheme following this logic, (see appendix I) and applied it to the synthetic datasets, (see algorithm 1). The scheme did improve the performance over the one-step prediction, and it outperformed even a fixed optimal $T^*$ for limit cycles. In more chaotic systems, the scheme performs worse than an optimal fixed temporal horizon, but it is important to notice that the optimal temporal horizon is not known, so the comparison is not fair to the iterative algorithm.

## 5.3   Loss landscape geometry for mechanistic models

Even though we focused on neural networks, AR models include mechanistic models such as differential equations, where the processes driving the dynamics of the system under study are known and already encoded in the model's structure, but where the parameters of the model are unknown. Here we extend the insights form our theory to this setting

We used the same systems as in the theory section, but this time we used the models described in appendix C, both to generate time series data and as models to train, where the parameters are unknown. We study the geometry of the forecasting loss from eq. (3) in fig. 6, where we changed a single parameter per model and evaluated the loss function at multiple temporal horizons. Notice that the roughness on the loss only appears for very high forecasting horizons $T$ with respect to the ones we saw in the previous sections. This happens because the models here are very similar to the models generating the data, thus it takes a much longer for the divergence to become high enough to affect training. A more complete view over all parameters is presented for the Lorenz system in appendix E.

Since the models that we train are the same as the ones used to generate the data, all models achieve a loss of zero when the parameters of the model are the same as those that generated the data. Thus, the generalization ratio $r(T_l, T_h)$ in theorem 4.6 is undetermined because both long and short $T$ provide a loss of zero, implying that a short $T$ generalizes as well as a long $T$ for the true optima. In contrast, we observe that the loss landscape becomes rougher as $T$ grows, in line with theorem 4.7, and for extremely large $T$ the loss is similar across almost all parameters, as per lemma A.5.

Thus, our theory would suggest that it is always better to train with the smallest possible $T$, when the system is known and perfectly deterministic. However, this statement should be contrasted with the practical limitations of mechanistic models, which are often not perfect, and real observations, which are contaminated with noise. Indeed, empirical studies considering noise and imperfect models suggest that there is a trade-off in the choice of $T$ similar to the one we found with MLPs (Pisarenko & Sornette, 2004; Boussange et al., 2024).

# 6 Conclusion and outlook

Our work establishes a fundamental connection between the temporal horizon $T$ of an autoregresive model and the geometry of the loss landscape. From a practical standpoint, we show that common training paradigms such as single step prediction or matching the testing horizon—are rarely optimal. Instead, the ideal $T$ depends on the system's intrinsic dynamics (e.g., Lyapunov exponent $\lambda$), but practical considerations like model capacity, number of epochs, learning rate and stochasticity also play a role. Although our theory considers Markovian systems, the general principles laid out should hold for more complex systems with long-term dependencies and associated architectures such as RNNs and Transformers.

Our findings motivate a principled approach to hyperparameter search, where $T$ is selected based on the system's dynamical properties, or updated during training. Future works should investigate this further, for example by jointly optimizing $T$ and the learning rate by leveraging loss landscape curvature, building on ideas from Roulet et al. (2024). Such an approach could extend beyond static horizons: many systems exhibit time-varying predictability (e.g., intermittent chaos or quasi-periodicity Zhou et al. (2006)), where a dynamic $T$ could sharpen the trade-off between learnability and generalization. For instance, shorter horizons might allow training on unpredictable regions, while longer horizons can refine learning in more ordered regions. This could unlock more efficient training for complex, non-stationary systems.

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

# A    Proofs for propositions and theorems

In the following subsections, we provide full proofs or sketches for all of the non-trivial mathematical claims made in the main text. Note that these subsections have been arranged so that the relevant theorems or propositions follow the same order as in the main text.

## A.1    Existence of neural network for epsilon-regions

**Lemma A.1.** *For any $\epsilon > 0$ and any real $p \in [0,1]$, there exists a number of observations $M_\epsilon$ and a sufficiently large neural network that has an $\epsilon$-bounded region with a probability $p$*

*Proof.* By ergodicity, for a sufficiently large $M$, the distribution of samples will converge to the stationary distribution. At some point there will be enough points in the (bounded) stationary distribution to guarantee that all points have a neighbour at a distance of $\epsilon$ with a probability larger than $p$. With a sufficiently large neural network, we can set the parameters such that every point $x$ in the neighborhood of $x(t)$ gets sent to $x(t+1)$. By construction, this will always map $f(x + \epsilon\vec{r}, \theta)$ onto either $f(x)$ or a close neighbour which will, by the compressive nature of the projection, fulfill the conditions of our definition. $\square$

**Lemma A.2.** *In a multi-layer perceptron with at least one hidden variable, the training loss that can be achieved decreases with the size of the hidden layers.*

*Proof.* Consider a multi-layer perceptron with at least one hidden layer and $N_l$ neurons per hidden layer and trajectory of length $M$ in a system with $D$ variables where the MLP achieves a training error per sample of $e(m) = \|x(m) - f(x(m-1), \theta)\|$. Then add $D + 1$ neurons to the first hidden layer and one neuron to any subsequent layers. We can create a bounding box with the extra $D + 1$ neurons for the sample $m_{\max}$ corresponding to the maximum $e(m_{\max})$ , and have one neuron in any subsequent layer whose only task is to transfer that signal to the output layer, correcting the error. $\square$

**Lemma A.3.** *For a stochastic dynamical system of the form $\dot{x} = \phi(x) + \xi$, where*

$$\|f(x + \epsilon\vec{r}, \theta) - (f(x, \theta) + J_\phi(x)\vec{r}\epsilon)\| < \epsilon^2 \tag{10}$$

*and $\xi$ is a random random vector such that $Pr\left[\|\xi\| > b\right] < \alpha$,*

$$Pr\left[\|f(x + \epsilon\vec{r}, \theta) - (f(x, \theta) + J_\phi(x)\vec{r}\epsilon)\| < \epsilon^2 + b\right] < \alpha \quad \forall x \in Conv(\mathbf{x}),$$

*Proof.* We are simply adding the noise term to the definition of $\epsilon$-bounded region, and note that since the noise is unrelated to the dynamics, the errors add up. $\square$

*Remark* A.4. If $b$ is small with $\alpha$ approaching one, then the system is effectively deterministic in the short term horizon.

**Lemma A.5.** *Consider a model that approximates a dynamical system that is not stable and has a bounded state space in its stationary distribution. For an initial position $x(0)$, the forecasting error as $T \to \infty$ behaves as a random variable with expected value*

$$E_{x(0)}\left[\|f^K(x(0)) - x(T|x(0))\|^2\right] = \begin{cases} 0 & \text{If the model perfectly predicts the system} \\ \mathcal{O}\left(Var_\mathcal{X}[x]\right) & \text{If the model is not strictly perfect} \end{cases} \tag{11}$$

*where $x(T|x(0))$ is the state of the system at time $T$ for a starting position $x(0)$ and $Var_\mathcal{X}[x]$ is the variance of the state space.*

*Proof.* The gist of our argument is that as the model and the system are ergodic, their difference is also ergodic (if they are different) or non-existent (if the model is perfect). Thus, for infinite time horizons the error in the forecasting follows a 0-1 law. If the model perfectly captures the dynamics, the error is clearly 0. If the model is not perfect at any region in the stationary distribution of the dynamical system, there will

be a perturbation of the trajectory. Such perturbations will not fade away since our system is not stable. Because the system is bounded but the trajectory is infinitely long, any region will be visited infinite times. Thus the error in the trajectory will accumulate. Eventually, any correlation between the model and the dynamical system will disappear. By ergodicity, the model and the system will generate points that are effectively independent with a variance of order $\text{Var}_{\mathcal{X}}[x]$. The expected value of the difference between two random variables is the sum of their variances, and since the model is similar to the system, the variances are within the same order of magnitude. □

## A.2 The model captures the dynamics

**Lemma A.6** (Approximate Jacobian). *For $\epsilon \to 0$, in an $\epsilon$-bounded region of the parameter space the Jacobian of the model converges to the Jacobian of the true dynamics*

$$\lim_{\epsilon \to 0} \|J_\phi(x) - J_f(x)\|_{OP} = 0 \tag{12}$$

*where $\|\|_{OP}$ is the absolute value of the largest eigenvalue of the matrix (also known as the operator norm).*

*Proof.* By using the equation in the definition of $\epsilon$-bounded region,

$$\left\| f(x + \sqrt{\epsilon}\vec{r}, \theta) - \left( f(x, \theta) + J_\phi(x)\vec{r}\sqrt{\epsilon} \right) \right\| = \left\| \left( f(x + \sqrt{\epsilon}\vec{r}, \theta) - f(x, \theta) \right) - J_\phi(x)\vec{r}\sqrt{\epsilon} \right\| < \epsilon$$

and taking the a Taylor expansion,

$$\left\| \left( f(x + \sqrt{\epsilon}\vec{r}, \theta) - f(x, \theta) \right) - J_\phi(x)\vec{r}\sqrt{\epsilon} \right\| \approx \| J_f(x)\vec{r}\sqrt{\epsilon} - J_\phi(x)\vec{r}\sqrt{\epsilon} \| < \epsilon$$

which converges to an equality in the limit $\epsilon \to 0$, yielding

$$\|J_f(x)\vec{r}\sqrt{\epsilon} - J_\phi(x)\vec{r}\sqrt{\epsilon}\| = \sqrt{\epsilon}\|J_f(x)\vec{r} - J_\phi(x)\vec{r}\| < \epsilon$$
$$\|J_f(x)\vec{r} - J_\phi(x)\vec{r}\| < \sqrt{\epsilon}$$

□

## A.3 Bounded loss

**Lemma A.7** (Bounded loss). *In an $\epsilon$-bounded region of the parameter space, the loss is bounded by*

$$\mathcal{L}(\theta, T) \leq \left( \max_{x_1, x_2 \in \mathcal{X}_0} \|x_1 - x_2\|_2^2 + 2\epsilon \right) = \mathcal{L}_\epsilon^{\max} \quad \forall T \tag{13}$$

*Proof.* For any dynamical system with a bounded state space, there is a maximum distance between its two farthest points. Since the model itself is also at a maximum distance from any point in the original system,

$$\mathcal{L}(\theta, T) \leq \max_{x_1, x_2 \in \mathcal{X}_\epsilon} \|x_1 - x_2\|_2^2 \leq \left( \max_{x_1, x_2 \in \mathcal{X}_0} \|x_1 - x_2\|_2^2 + 2\epsilon \right) = \mathcal{L}_{\max, \epsilon} \quad \forall T \tag{14}$$

Note that for $\epsilon \to 0$, the upper bound is the variance of the ergodic distribution of the system. □

## A.4 Relating gradients and temporal horizons

**Theorem A.8** (Unbounded loss gradient). *Consider a dynamical system which is either chaotic, contains locally unstable trajectories or contains limit cycles, and a corresponding model with an $\epsilon$-bounded region of the parameter space $\Theta_\epsilon$ with non-zero loss. When the forecasting horizon $T$ is large, the expected gradient magnitude grows with $T$ as*

$$\frac{\|\nabla_\theta \mathcal{L}(\theta, T)\|}{\|\nabla_\theta \mathcal{L}(\theta, 1)\|} = \begin{cases} \mathcal{O}(e^{\lambda T}), & \text{for chaotic systems} \\ \mathcal{O}(\omega T), & \text{for limit cycles} \end{cases} \tag{15}$$

*where $\|\cdot\|$ is the Euclidean norm.*

*Proof.* We start by analyzing the loss given in eq. (3), which is a function of the parameters $\theta$ given the true trajectory at any time $t$ by $x(t)$. To understand how learning operates under this loss function, we study how the landscape induced by $\mathcal{L}(\theta, T)$ varies under different choices of $T$. We can get the gradient of eq. (3), our loss function, by evaluating

$$\nabla_\theta \mathcal{L}(\theta, T) = \frac{1}{M - K} \sum_{k=1}^{M-K} \sum_{\tau=1}^{K} \frac{\partial f^\tau(x, \theta)}{\partial \theta} \bigg|_{x(k), \theta} (x(k+\tau) - f^\tau(x(k), \theta)), \tag{16}$$

where we must still compute the partial derivatives of the function $f^\tau$. For $\tau = 1$, we denote the derivatives of $f^1 = f(x, \theta)$ as

$$J_\theta(x, \theta) = \nabla_\theta f(x, \theta), \quad J_x(x, \theta) = \nabla_x f(x, \theta) \tag{17}$$

Using the chain rule, we obtain the expression

$$J_\theta^\tau(x, \theta) = \nabla_\theta f^\tau(x, \theta) = \sum_{l=1}^{\tau} J_x^l(f^{\tau-l}(x), \theta) J_\theta\left(f^l(x), \theta\right), \tag{18}$$

where $J_x^k(x, \theta) = \prod_{k=1}^{\tau} J_x(f^k(x), \theta)$. To analyze eq. (16) and thus eq. (3), we can use some knowledge we have about the properties of the state space and parameter space Jacobians involved in eq. (18). Specifically, the state space Jacobian, $J_x(x, \theta)$, is directly dependent on the trajectories of the system and indirectly dependent on the parameters. The notable implication of this is that any set of parameters where the model captures the dynamics reasonably well, the Jacobian in the state space of the model will be very closely aligned with the Jacobian of the system dynamics, which is given by the data (see Lemma A.6). In contrast, the Jacobian of the parameters $J_\theta(x, \theta)$ depends on the model and its parametrization.

We can therefore compute the ratio of gradients, $\dfrac{\|\nabla_\theta \mathcal{L}(\theta, T)\|}{\|\nabla_\theta \mathcal{L}(\theta, 1)\|}$ for a given $\theta$ by using

$$\|\nabla_\theta \mathcal{L}(\theta, T)\| = \frac{1}{M - K} \sum_{m=1}^{M-K} \frac{1}{K} \sum_{k=1}^{K} \sum_{l=1}^{k} J_x^l(f^{k-l}(x), \theta) J_\theta\left(f^l(x), \theta\right) (x(m+k) - f^k(x(m), \theta)). \tag{19}$$

In chaotic systems, we know that the product of Jacobians scales as

$$\lim_{\tau \to \infty} \frac{1}{\tau} \ln\left(\prod_{l=1}^{\tau} J_x(f^l(x), \theta)\right) = \lambda T/K \tag{20}$$

where $\lambda$ is the Lyapunov exponent of the system, which is positive (Wolf et al., 1985), and $T/K$ is simply a scaling factor to convert the forecasting steps into the timescale of the system. Thus, loss behaves as

$$\|\nabla_\theta \mathcal{L}(\theta, T)\| = \mathcal{O}\left(e^{\lambda T}\right). \tag{21}$$

If we apply the same logic to the gradient with a time horizon of one (or any other reference length),

$$\frac{\|\nabla_\theta \mathcal{L}(\theta, T)\|}{\|\nabla_\theta \mathcal{L}(\theta, 1)\|} = \mathcal{O}(e^{\lambda(T-1)}) = \mathcal{O}(e^{\lambda T}) \tag{22}$$

We also analyze limit cycles by focusing on the phase of the rotation. A trajectory produced by $f^\tau(x, \theta)$ where the period is not exactly the same as in the original dynamical system will slowly drift as the phases will change move at slightly different angular velocities. In more formal terms, if we parameterize the limit cycle by the phase $\varphi$ and ignore its support we obtain

$$J_\theta^\tau(\varphi, \theta) = \sum_{l=1}^{\tau} \left[\prod_{m=l+1}^{\tau} J_\varphi(f^m(x), \theta)\right] J_\theta\left((f^l(\varphi), \theta\right) = \sum_{l=1}^{\tau} J_\theta\left(f^l(\varphi), \theta\right) \tag{23}$$

where the position is repeated for large $\tau$ (this is a limit cycle), with a difference given by the quality of the approximation. To make this simple, we can consider the growth per cycle, and count the cycles,

$$J_\theta^\tau(\varphi, \theta) \approx c \int_0^{2\pi} J_\theta(\varphi, \theta) \, d\varphi = \tau\omega \int_0^1 J_\theta(a, \theta) \, da \tag{24}$$

where the integral gives the average Jacobian through the cylce, and $c = \frac{\tau}{p} = \frac{1}{2\pi}\tau\omega$ is the number of cycles (notice the change of variable in the integral).

As such with chaotic behavior and locally unstable trajectories, the norm of the Jacobian grows exponentially with $T$. For limit cycles, we know that the Jacobian grows at least linearly with $\omega T$.

$\square$

**Corollary A.9.** *For the models considered in A.8, the hessian of the model around its minima grows as*

$$\frac{\|H(\theta, T)\|^*}{\|H(\theta, 1)\|^*} = \begin{cases} \mathcal{O}(e^{\lambda T}), & \text{for chaotic systems} \\ \mathcal{O}(\omega T), & \text{for limit cycles} \end{cases} \tag{25}$$

*where $\|\cdot\|^*$ is the Nuclear norm.*

*Proof.* The gist of the proof is to show that as the gradient grows with $T$, the second derivative must also grow to keep up. We build this connection between the gradient and the Hessian through the divergence theorem.

Consider the gradient of the loss $\nabla\mathcal{L}(\theta, T)$ as a vector field in the space of parameters. Pick a minima of the loss $\theta_T^{\min}$, and a ball around it denoted by $B_{\theta_T^{\min}}$ and its spherical boundary $S_{\theta_T^{\min}}$. Then, by the divergence theorem,

$$\int_{B_{\theta_T^{\min}}} \mathbf{div}\nabla\mathcal{L}(\theta, T) dV = \int_{S_{\theta_T^{\min}}} \nabla\mathcal{L}(\theta, T) d\vec{S} \tag{26}$$

where $d\vec{S}$ is a vector normal to the sphere scaled by a differential surface element, and $dV$ a differential element of volume. Since $\theta_T^{\min}$ is a minima, for a small enough ball the gradient of the loss is always pointing towards the inside, hence the product with $d\vec{S}$ will have a constant sign. Thus,

$$\int_{B_{\theta_T^{\min}}} \mathbf{div}\nabla\mathcal{L}(\theta, T) dV = \int_{S_{\theta_T^{\min}}} \|\nabla\mathcal{L}(\theta, T) d\vec{S}\|. \tag{27}$$

Now, note that the divergence of the gradient is the trace of the Hessian,

$$\int_{B_{\theta_T^{\min}}} \mathbf{Tr}\left[H(\theta, T)\right] dV = \int_{S_{\theta_T^{\min}}} \|\nabla\mathcal{L}(\theta, T) d\vec{S}\|. \tag{28}$$

Now we simply need to remember that the ball was relatively small and the loss smooth, so that we can use the approximation

$$\mathbf{Tr}\left[H(\theta, T)\right] \approx \mathbf{Tr}\left[H(\theta_T^{\min}, T)\right] \quad \forall\theta \in B_{\theta_T^{\min}}, \tag{29}$$

and noticing that the trace is sum of eigenvalues of a matrix, which is the nuclear norm,

$$\mathbf{Tr}\left[H(\theta_T^{\min}, T)\right] \int_{B_{\theta_T^{\min}}} dV = \|H(\theta_T^{\min}, T)\|^* \int_{B_{\theta_T^{\min}}} dV = \int_{S_{\theta_T^{\min}}} \|\nabla\mathcal{L}(\theta, T) d\vec{S}\|. \tag{30}$$

and if we apply this for $T = 1$ and divide,

$$\frac{\|H(\theta_T^{\min}, T)\|^*}{\|H(\theta_1^{\min}, 1)\|^*} = \frac{\int_{S_{\theta_T^{\min}}} \|\nabla\mathcal{L}(\theta, T) d\vec{S}\|}{\int_{S_{\theta_1^{\min}}} \|\nabla\mathcal{L}(\theta, 1) d\vec{S}\|} = \begin{cases} \mathcal{O}(e^{\lambda T}), & \text{for chaotic systems} \\ \mathcal{O}(\omega T), & \text{for limit cycles} \end{cases}, \tag{31}$$

where the last step comes from proposition 4.3,

$\square$

## A.5 Minima with longer forecasting horizons generalize better

**Theorem A.10** (Minima with long forecasting horizons generalize to lower horizons). *Consider a model $f(\cdot, \theta)$ with $\theta$ in an $\epsilon$-bounded region. We assume the existence of two minima $\theta_l^{\min}$ and $\theta_h^{\min}$ for the losses $\mathcal{L}_\mathbf{x}(\theta, T_l)$ and $\mathcal{L}_\mathbf{x}(\theta, T_h)$ respectively with $T_h > T_l$ such that $\theta_l^{\min}$ would converge to $\theta_h^{\min}$ if trained on $T_h$, and vice-versa. Then the difference in the change in losses follows*

$$r(T_h, T_l) = \begin{cases} \mathcal{O}(e^{\lambda(T_h - T_l)}), & \text{for chaotic/unstable} \\ \mathcal{O}(\omega(T_h - T_l)), & \text{for limit cycles} \end{cases} \tag{32}$$

*Proof.* Consider the path $p$ of the gradient between the minima $\theta_T^{\min}$ for the temporal horizon $T$ and another point $\theta^{\text{basin}}$ such that it will converge to $\theta_T^{\min}$ through training by gradient descent. On this path we can compute the change in loss,

$$\mathcal{L}(\theta_T^{\min}, T) - \mathcal{L}(\theta^{\text{basin}}, T) = \int_{\theta \in p} \langle \nabla \mathcal{L}(\theta, T), \vec{u}(\theta, p) \rangle d\theta = \int_{\theta \in p} \|\nabla \mathcal{L}(\theta, T)\| d\theta \tag{33}$$

$$= \int_{\theta \in p} \|\nabla \mathcal{L}(\theta, 1)\| \frac{\|\nabla \mathcal{L}(\theta, T)\|}{\|\nabla \mathcal{L}(\theta, 1)\|} d\theta \tag{34}$$

where $\vec{u}(\theta, p)$ is the unit vector on the direction of the path of the gradient, and since the path is defined by the gradient itself, $\langle \nabla \mathcal{L}(\theta, T), \vec{u}(\theta, p) \rangle = \|\nabla \mathcal{L}(\theta, T)\|$. By proposition 4.3, the ratio of loss norms in the integral either grows exponentially or linearly with $T$. By substituting $T = T_h$ and $T_l$ then taking the fraction in eq. (32) we complete the proof. □

## A.6 Loss landscape roughness

**Lemma A.11.** *Consider a continuous differentiable function $g(s)$ on an interval $s \in [s_a, s_b]$ bounded from above and below,*

$$g_{\min} < g(s) < g_{\max} \tag{35}$$

*with a variation in value*

$$v_g = \int_{z_a}^{z_b} \left| \frac{dg(s)}{dz} \right| ds > 2n(g_{\max} - g_{\min}), \quad n \in \mathbb{N}^+. \tag{36}$$

*then $g(s)$ has at least $n$ minima and $n$ maxima in the interval considered.*

*Proof.* Assume without loss of generality that $g$ is initially increasing. That is, $\frac{dg(s_a)}{ds} > 0$. Further define,

$$v_g(s) = \int_{s_a}^{s} \left| \frac{dg(s)}{ds} \right| ds \tag{37}$$

which we know to be a semi-positive monotonically increasing function with range $[0, U]$ where $U > 2n(g_{\max} - g_{\min})$. Consider the first point $z_1$ such that $v_g(s_1) > g_{\max} - g_{\min}$. Assume for contradiction that $g$ does not have a maxima on the interval $[s_a, s_1]$. We know that,

$$g(s_1) = g(s_a) + \int_{s_a}^{s_1} \frac{dg(s')}{ds'} ds'. \tag{38}$$

By assumption, $\frac{dg(s')}{ds'}$ is semi-positive and must not change sign on the interval. As such,

$$g(s_1) = g(s_a) + \int_{s_a}^{s_1} \left| \frac{dg(s')}{ds'} \right| ds' > g(s_a) + g_{\max} - g_{\min} \Rightarrow g(s_1) - g(s_a) > g_{\max} - g_{\min}. \tag{39}$$

This implies that $g(s_1) - g(s_a) > g_{\max} - g_{\min}$, which is a contradiction. Therefore on the interval $[s_a, s_1]$ there must exist a maximum $s_1^*$. Applying the same logic to the interval $[s_1^*, s_2]$ such that $v_g(s_2) > 2(g_{\max} - g_{\min})$ proves there must be a minimum on that interval. Further applying these two arguments $n - 1$ more times shows $g$ must have at least $n$ minima and $n$ maxima on $[s_a, s_b]$. □

**Theorem A.12** (Loss landscape roughness)**.** *For any two points $\theta_1, \theta_2$ in an $\epsilon$-bounded region of the parameter space that are not in a connected region of zero loss[2], the number of minima and maxima along the line segment that connects them will grow as*

$$z(T) = \begin{cases} \mathcal{O}(e^{\lambda T}\|\theta_1 - \theta_2\|), & \text{for chaotic or unstable systems} \\ \mathcal{O}(T\|\theta_1 - \theta_2\|), & \text{for limit cycles} \end{cases} \tag{40}$$

*Proof.* Let us look at arbitrary elements of the parameter space $\theta_1, \theta_2 \in \Theta$. By lemma A.7, the difference in their loss is bounded,

$$|\mathcal{L}(\theta_1, T) - \mathcal{L}(\theta_2, T)| \le \mathcal{L}_\epsilon^{\max}. \tag{41}$$

We now consider the variation in the loss function that goes from $\theta_1$ to $\theta_2$ through a straight line,

$$v_T(\theta_1, \theta_2) = \int_{l=\theta_1}^{\theta_2} \left| \frac{\partial \mathcal{L}(l, T)}{\partial l} \right| dl, \tag{42}$$

where $l$ is the variable that represents points in a line from $\theta_1$ to $\theta_2$. We note that the derivative of the loss is a projection of the gradient onto the line,

$$\frac{\partial \mathcal{L}(l, T)}{\partial l} = \langle \nabla_\theta \mathcal{L}(l, T), \vec{u}_{\theta_1 \to \theta_2} \rangle \tag{43}$$

where $\vec{u}_{\theta_1 \to \theta_2}$ is the unitary vector of the line going from $\theta_1$ to $\theta_2$. By proposition 4.3, the value of this projection grows exponentially in a chaotic or locally unstable trajectory. In a limit cycle, the growth is or linearly with $T$. Thus, we only need linear growth in eq. (43) for an unbounded value of $v_T(\theta_1, \theta_2)$. Taking the integral over the line,

$$\mathcal{O}\left[ \int_{l=\theta_1}^{\theta_2} \left| \frac{\partial \mathcal{L}(l, T)}{\partial l} \right| dl \right] = \begin{cases} e^{\lambda T}\|\theta_1 - \theta_2\|, & \text{for chaotic or unstable systems} \\ T\|\theta_1 - \theta_2\|, & \text{for limit cycles} \end{cases} \tag{44}$$

and by lemma A.11, we then have that $z(T)$ must grow with lower bound given by eq. (44), thus completing the proof.

$\square$

Before continuing we need to introduce the notion of box counting dimension (also known as Minkowski–Bouligand dimension), which is defined as

$$\dim_{\text{box}}(\mathcal{L}(\theta, T)) := \lim_{\varepsilon \to 0} \frac{\log N(\varepsilon)}{\log(\varepsilon^{-1})} \tag{45}$$

where $N(\varepsilon)$ is the number of balls of radius $\varepsilon$ needed to cover a surface (here the loss). In smooth surfaces, the box-counting dimension is equal to the topological dimension (the number of dimensions within the manifold), and the number of balls would scale with the number of parameters. However, if the box counting dimension is higher than its topological dimension, the surface is a fractal Mandelbrot (1983); Edgar & Edgar (2008). We will use this in the next corollary, but it is also a relevant concept in dynamical systems, for example in Taken's theorem which we will use later.

**Corollary A.13** (The loss function is a fractal)**.** *For an $\epsilon$-bounded region of the parameter space, in the limit $T \to \infty$, the surface of the loss is a fractal occupying a volume that converges to*

$$|\Theta_\epsilon|\sqrt{\frac{Var[x \in \mathcal{X}]}{M}} < V < |\Theta_\epsilon|\sqrt{\frac{Var[x \in \mathcal{X}] + \epsilon^2}{M}} \tag{46}$$

*where $|\Theta_\epsilon|$ is the area of the $\epsilon$-bounded region and $Var[x \in \mathcal{X}]$ is the variance of the invariant probability distribution in the dynamical system.*

---

[2]This case is not covered in the assumptions on proposition 4.3, but if we are in such a region there is no learning

*Proof.* The topological dimension is the number of parameters in the loss landscape. For the box counting dimension, we consider a cover of the parameter space by infinitesimally small open balls. For each one of those balls, however, the loss function has unbounded gradients and thus we can always have a gradient large enough so that a ball of a radius $\epsilon$ would not cover the neighborhood of a point. Hence, we would need the number of parameters plus one dimensions to cover the loss landscape surface. From the dynamics perspective, any parameter setting where $\mathcal{L}(\theta, 1) > 0$ implies that there is at least some cases where the model deviates from the system. For some high value of $t^*$, the deviations accumulate until the position of the model is uncorrelated with the position of the dynamical system observed as $x(t^*)$. If $T \gg t^*$, the observations are uncorrelated with the model, and thus we are sampling a variable with variance $\text{Var}[x \in \mathcal{X}]$. We have $M$ samples and a margin of error of $\epsilon$, which gives us the term in the square root. The multiplication by $\Theta_\epsilon$ is simply because we need to consider all the parameter sets. Note that even if some parameters result in a zero-loss, they form surfaces of lower dimensionality and thus have zero volume. $\square$

### A.7 Effects of noise

Our theory builds on the assumption that the $\epsilon$ in proposition 4.3 is small enough so that models with $\theta \in \Theta_\epsilon$ will approximate the system dynamics well (lemma A.6). In stochastic systems the $\epsilon$ is bounded away from zero, meaning that in large temporal horizons any model is effectively a random guess (A.5). However, there is often a temporal horizon in which the noise is still small enough so that predictions are possible (A.14)

**Proposition A.14** (Temporal horizon limits on systems with noise)**.** *In a dynamical system whose dynamics are given by limit cycle or chaos, and where there is an extra term inducing random noise with a variance uniform in all dimensions and with value $\sigma^2$ (per dimension), there is a maximum temporal horizon $T_{\max}$ after which forecasting is effectively a random guess given by*

$$
T_{\max} = \begin{cases} \mathcal{O}\left(\dfrac{\ln(S) - \ln(\sigma)}{\lambda}\right), & \text{for chaotic or unstable systems} \\ \mathcal{O}\left(\dfrac{L}{\sigma}\right), & \text{for limit cycles} \end{cases} \tag{47}
$$

*where $L$ is the length of the limit cycle and $S$ the maximum radius of the state space in the chaotic system.*

*Proof.* The gist is that in any dynamical system where the noise accumulates, at some point the contribution of the noise to the current state of the system is going to be as large as the contribution of the deterministic dynamics. For a chaotic system a perturbation of size $\sigma$ will scale exponentially as $\sigma e^{\lambda T}$, and when $\sigma e^{\lambda T} > S$, the effect of that perturbation is as large as the state space. For a limit cycle we can apply the same logic, except that the noise grows as $T\sigma$ and we use the length of the limit cycle. $\square$

### A.8 Generalization to partially observable systems and recurrent networks

We develop our theory using mostly autoregressive neural networks in systems with full observability. However, we can extend it to the more general settings of non-fully observable models. However, this requires either recurrent networks or models with a context window. In the following, we prove that any neural network that forecasts with high precision needs to have access to a representation of the state of the dynamical system which can be mapped one to one to all the states of the system as if it was fully observed. In the dynamical systems literature, this is usually done by using a higher-dimensional representation of the system, which is known as an embedding. Then we show that enhancing feed-forward networks with hidden states or context can provide that representation. We note that we don't believe that those results are new, but we could not find a direct reference, so we add it here for completeness.

The first step is to prove that an arbitrarily small forecast requires a representation of the state of the system that captures it fully. This is effectively the converse of A.6.

**Lemma A.15.** *Consider a smooth deterministic, ergodic, discrete dynamical system, defined by $x(m) = \phi(x(m-1))$, an observation function $h \in \mathbb{R}^d$ and a function $f$ that forecasts it with a precision of $\epsilon \geq \|f(h(x)) - \varphi(x)\|$. For a sufficiently small $\epsilon$, $d$ must be higher or equal than the topological dimension of the dynamical system, and in the limit of $\epsilon = 0$, $h(x)$ must be an embedding of $x$.*

*Proof.* $f$ maps the potential precursors of each point to a prediction that is at a distance of $\epsilon$. We can thus define a partition of the dynamical system, where each point is in a ball of radius $\epsilon$ that contains its prediction. As $\epsilon$ goes to zero, this partition gives a unique value to each point in the system. To do so, the topological dimensions need to match. The fact that this converges to an embedding in the limit of small $\epsilon$ is straightforward, since every point corresponds to one image of $h$. $\square$

This proves that such architectures need that representation. In the next steps we prove that context windows or recurrent connections in the hidden layer are sufficient to do that.

The gist of our proofs relies on having a neural network that reconstructs the original dynamical system from observations of the current and past (partial) observations of the system, following Taken's theorem. This can then be replaced by a recurrent neural network using a construction that is presented in Deco & Schürmann (2012). Note that for simplicity we will assume a single observable variable per system, but the results are valid for more.

**Lemma A.16.** *Consider a deterministic dynamical system sampled at fixed time intervals, so that $x(m) = \phi(x(m-1))$ and an observation function $o(x) \in \mathbb{R}$ that is smooth and coupled to every dimension of $x$ such that $\frac{\partial o(x(m))}{\partial x_i(m-k)} \neq 0$ for some k. If there exists a neural network $f(\cdot, \theta)$ such that*

$$\|f^K(x(m), \theta) - x(m+K)\|^2 < \epsilon_f \tag{48}$$

*then for every error $\epsilon_f < \epsilon_g$ there exists a neural network $g(\cdot, \kappa)$ with parameters $\kappa$ and input dimension $d \leq 2b+1$ where b is the box counting dimension of the invariant manifold of x such that*

$$\|g^T([o(x(m)), o(x(m-1)), o(x(m-2))...o(x(m-k))], \kappa) - o(x(m+K))\|^2 < \epsilon_g. \tag{49}$$

*Proof.* Note that the conditions on the dynamical system and the observation function are precisely those of the Taken's theorem Takens (1981a), which states that there is an embedding of $[o(x(m)), o(x(m-1)), o(x(m-2))...o(x(m-k))]$ that captures the full attractor of the original dynamical system. As a consequence of the embedding, we can construct a smooth function $\nu$ which takes $[o(x(m)), o(x(m-1)), o(x(m-2))...o(x(m-k))]$ and maps it to $x(m)$.

A function such as $\nu$ can be approximated with arbitrary precision by a neural network $h(\cdot, \iota)$ with parameters $\iota$, by the universal approximation theorem.

Thus, we can use h to recover x from $o(x)$, and then compose the results with the original neural network $f$, giving us $g(\cdot, \kappa) = f(h([o(x(m)), o(x(m-1)), o(x(m-2))...o(x(m-k))], \iota), \theta)$ which is a composition of the reconstruction and the original $f$. The error on each timestep will be a composition of the errors induced by h and $f$. $\square$

*Remark* A.17. Note that the transformer architecture fulfills the conditions of the theorem, but the self-attention mechanism is known to disregard the temporal information that is crucial for forecasting Zeng et al. (2023). The current solution is the PatchTST model, which effectively uses this embedding (Nie et al., 2022).

**Lemma A.18.** *Consider a discrete deterministic dynamical system defined by $x(m) = \phi(x(m-1))$ and an observation function $o(x) \in \mathbb{R}$ that is smooth and coupled to every dimension of x. If there exists a neural network $f(\cdot, \theta)$ such that*

$$\|f^K(x(m), \theta) - x(m+K)\|^2 < \epsilon_f, \tag{50}$$

*then for all $\epsilon_g > \epsilon_f$ there exist a recurrent neural network $g(\cdot, h, \kappa)$ where h is the hidden state of dimension $d \leq 2b+1$ with b being the box counting dimension of the invariant manifold of x such that if the inputs to g have been $[o(x(m)), o(x(m-1)), o(x(m-2))...o(x(m-k))]$ then*

$$\|g^T(o(x(m)), h, \kappa) - o(x(m+T))\|^2 < \epsilon_g. \tag{51}$$

*Proof.* We use a similar argument as presented in (Deco & Schürmann, 2012). We build a hidden layer with recurrent connections $h_i \rightarrow h_{i+1}$, and the input $o(x) \rightarrow h_1$. Then the hidden states save the last $d$ inputs, and we can apply lemma A.16. $\square$

*Remark* A.19. Note that the construction simply proves the existence of such a network, not that it needs to be build in this manner. Other architectures like vanilla recurrent neural networks or long-short term memory networks could implement this construction, but it does not need to be the best solution.

Now we put together lemma A.15, lemma A.16, and lemma A.18.

**Proposition A.20.** *Assume the existence of a smooth neural network that forecasts a deterministic dynamical system with precision $\epsilon$ from a partial observation of the state $o(x)$ and a hidden state or context window $h(x)$. For a sufficiently small $\epsilon$, $[o(x), h(x)]$ is an embedding of the dynamical system, for which we can define the functions $g, e$ such that*

$$e(x(m+\tau)) = g \circ \cdots \circ g\left(e(x(m)), \theta\right) = g^\tau\left(e(x(m)), \theta\right), \tag{52}$$

*where $e(x)$ is an embedding of $x$.*

*Proof.* Lemma lemma A.15 makes the embedding necessary, and lemmas A.16 and A.18 prove that RNNs and transformers can have it (respectively). $g$ is then simply a function equivalent to $f$ in eq. (2), but applied to the embedding $e$. $\qquad\square$

*Remark* A.21. Rewriting our proofs with the extended state space would be cumbersome, because the loss would only refer to part of the state space. We would have to then create a surrogate loss covering the extended space, and bound it to the loss on the observed space. This would include extra definitions and clarifications for each theorem, and compromise readability.

## B   Necessary assumptions

Some of the assumptions in our theory are necessary to have any form of statistical learning. Here we explain why.

**Stationarity**: In dynamical systems, stationarity refers to the property where the statistical characteristics of a system's behavior remain constant over time. This means that the system's "behavior" or patterns don't change, regardless of when you observe them. It is impossible to make forecasts based on statistical learning if the statistics of the system are not known (or change). Thus, this assumption is unavoidable.

**Ergodicity**: In dynamical systems, ergodicity describes a system's behavior where the time average of a property along a trajectory is equal to the space average of that property. Essentially, a system is ergodic if a single trajectory visits all parts of the system's phase space (or a significant portion of it) in a statistically representative way over a long period. Since we need to be able to apply statistical learning, some form of ergodicity is necessary to guarantee that the data seen during training is also representative of the data during testing.

**Uniform state coverage**: The last critical assumption is that the training data should capture all the relevant state space seen during testing. It is in principle impossible to prove forecasting capacity in states that have never been observed before without prior knowledge on the dynamics of the system, because one could always imagine some extremely convoluted dynamics happening only on that unobserved region of the state space. If we have some prior information though, we might be able to leverage it. In the extreme, this is corresponds to mechanistic models where the physics of the system are known, but the specific parameters are not. Note that we cover this setting in section 5.3.

Finally, note that it is possible to partially relax those assumptions, for example by having quasi-stationary systems or mesastability. However, this makes proofs significantly more cumbersome, and introduces subtle mathematical notions that are not easy to asses.

## C   Dynamical systems equations

Here we describe the differential equations governing the four dynamical systems used in the main text.

### C.1   Lorenz attractor

The Lorenz attractor is a three dimensional dynamical system described by the following equations

$$\frac{\mathrm{d}x}{\mathrm{d}t} = \sigma(y - x), \quad \frac{\mathrm{d}y}{\mathrm{d}t} = x(\rho - z) - y, \quad \frac{\mathrm{d}z}{\mathrm{d}t} = xy - \beta z \tag{53}$$

where we take $\sigma = 10.0$, $\rho = 28.0$ and $\beta = \frac{8}{3}$. The system was originally described by Edward Lorenz and is a simplification of hydrodynamic flows. We note that its trajectories are bounded and non-periodic (Lorenz, 1963).

### C.2   Double pendulum

The double pendulum is a simple mechanical system that exhibits chaotic behavior Gupta et al. (2014). It consists of two masses rigidly linked to each other, both can rotate freely around their anchor point. The first mass is connected to a center point and the second mass is linked to the first mass.

The dynamics are determined by applying the standard equations of motion from Newtonian mechanics, resulting in eq. (54)ff.

$$x_1 = \frac{l}{2} \sin \theta_1 \tag{54}$$

$$y_1 = -\frac{l}{2} \cos \theta_1 \tag{55}$$

$$x_2 = x_1 + l_2 \sin \theta_2 \tag{56}$$

$$y_2 = y_1 - l_2 \cos \theta_2 \tag{57}$$

### C.3   Food web model

We consider a generalized 7-species food-web model inspired by Hastings (1991); McCann & Yodzis (1994); Klebanoff & Hastings (1994); Post et al. (2000); Åkesson et al. (2021), where species interact through trophic and competitive relationships.

Denoting the abundance of species $i$ by $N_i$, and defining $N = (N_1, \ldots, N_7)$, the generalized community model can be expressed as

$$\frac{1}{N_i} \frac{d}{dt} N_i = r_i(u(t)) - \sum_{j=1}^{7} N_j \left[ \alpha_{i,j} + [F(N)]_{i,j} - [F(N)]_{j,i} \right]. \tag{58}$$

In eq. (58), the first two terms capture intrinsic growth rate and intra- and interspecific competition. The last two terms capture growth and loss due to trophic interactions.

The per capita growth rate $r_i$ may depend on a time-dependent environmental forcing $u(t)$. The competition coefficient between species $i$ and $j$ is denoted by $\alpha_{i,j}$. The feeding rate $[F(N)]_{i,j}$ of species $i$ on $j$ follows a functional response of type II

$$[F(N)]_{i,j} = \frac{q_{i,j} W_{i,j}}{1 + q_{i,j} H_{i,j} \sum_{k=1}^{7} W_{i,k} N_k}, \tag{59}$$

where $q_{i,j}$ and $H_{i,j}$ represent the attack rate and handling time of species $i$ when feeding on species $j$, respectively. We used the biologically realistic parameter values with $r(u(t))$ being constant. For $\omega = 0.2$, our system is defined by

$$
r = \begin{pmatrix} 1 \\ -0.15 \\ -0.08 \\ 1.0 \\ -0.15 \\ -0.01 \\ -0.005 \end{pmatrix}, \alpha = \begin{pmatrix} 1.0 & 0 & 0 & 0 & 0 & 0 & 0 \\ 0 & 0 & 0 & 0 & 0 & 0 & 0 \\ 0 & 0 & 0 & 0 & 0 & 0 & 0 \\ 0 & 0 & 0 & 1.0 & 0 & 0 & 0 \\ 0 & 0 & 0 & 0 & 0 & 0 & 0 \\ 0 & 0 & 0 & 0 & 0 & 0 & 0 \\ 0 & 0 & 0 & 0 & 0 & 0 & 0 \end{pmatrix}, W = \begin{pmatrix} 0 & 0 & 0 & 0 & 0 & 0 & 0 \\ 1 & 0 & 0 & 0 & 0 & 0 & 0 \\ 0 & \omega & 0 & 0 & 1 & 0 & 0 \\ 0 & 0 & 0 & 0 & 0 & 0 & 0 \\ 0 & 0 & 0 & 1-\omega & 0 & 0 & 0 \\ 0 & 0 & 1 & 0 & 0 & 0 & 0 \\ 0 & 0 & 0 & 0 & 0 & 1 & 0 \end{pmatrix},
$$

$$
H = \begin{pmatrix} 0 & 0 & 0 & 0 & 0 & 0 & 0 \\ 2.89855 & 0 & 0 & 0 & 0 & 0 & 0 \\ 0 & 7.35294 & 0 & 0 & 7.35294 & 0 & 0 \\ 0 & 0 & 0 & 0 & 0 & 0 & 0 \\ 0 & 0 & 0 & 2.89855 & 0 & 0 & 0 \\ 0 & 0 & 8.0 & 0 & 0 & 0 & 0 \\ 0 & 0 & 0 & 0 & 0 & 12.0 & 0 \end{pmatrix},
$$

$$
q = \begin{pmatrix} 0 & 0 & 0 & 0 & 0 & 0 & 0 \\ 1.38 & 0 & 0 & 0 & 0 & 0 & 0 \\ 0 & 0.272 & 0 & 0 & 0.272 & 0 & 0 \\ 0 & 0 & 0 & 0 & 0 & 0 & 0 \\ 0 & 0 & 0 & 1.38 & 0 & 0 & 0 \\ 0 & 0 & 0.1 & 0 & 0 & 0 & 0 \\ 0 & 0 & 0 & 0 & 0 & 0.05 & 0 \end{pmatrix}.
$$

The dynamics of the system are chaotic for this set of parameter values, but in short timescales they resemble a limit cycle. All non-zero coefficients were set as free parameters, excluding the coefficients of the adjacency matrix $W$, where only $\omega$ was considered as a free parameter.

### C.4 Limit cycle

The basic limit cycle system is defined below,

$$
\frac{1}{a}\frac{dx}{dt} = \mu x - y - x(x^2 + y^2), \quad \frac{1}{a}\frac{dy}{dt} = x + \mu y - x(x^2 + y^2) \tag{60}
$$

with parameters $a = 1.0, \mu = 0.4$.

### C.5 Implementation Details

The exact way in which the dynamical systems were solved depended on the figure. For fig. 6 they were solved in Julia using `ChaoticInference`Marti (2024) whereas for the other plots they were solved in Python using `diffrax` (Kidger, 2021).

#### C.5.1 Fitting dynamical systems

The four systems, Lorenz, double pendulum, ecological model and the limit cycle are implemented as dynamical systems, where an ODE gives rise to a trajectory.

As the dynamical systems are solved by numerical integration over some time and not autoregressive, an adjusted loss function (Equation 61) is used.

$$
\mathcal{L}_{\mathbf{x}}(\theta, T) = \sum_{m=0}^{\frac{M-T}{T}} \sum_{\tau=1}^{T} \| x(mT + \tau) - F^{\tau}(x(mT), \theta) \|. \tag{61}
$$

Where $F^\tau(x_0, \dots)$ denotes the numerical solution for $x(\tau)$.

Heavy lifting is done by the `ChaoticInference` library (Marti, 2024), namely data generation, loss evaluation and plotting. Data generation was done with the fully implicit, fifth order `RadauIIA5` solver(Rackauckas & Nie, 2017). For more details regarding initial conditions, timespan, etc. see the respective implementations.

### C.5.2 Fitting neural networks

The four systems were solved using the `diffrax` package developed by Kidger (2021) with 10 initial conditions sampled from a Gaussian distribution. For the dataset associated with the food web, we sample the system every 2 seconds, for those associated with the double pendulum they were sampled every 0.005 seconds, for the Lorenz they were either sampled every 0.005 or 0.04 depending on the experiment and for the limit cycle they were sampled every 0.5.

### C.6 Timestep choice

We selected the sampling timestep in such a way that we could observe the growths of the gradients within a reasonable range of $T$. Having a higher sampling rate would require computing a lot of values of $T$ and overburden our plots, while a too low $T$ would prevent the model from learning.

# D Geo-spatial datasets

In the following section, we expand on the geo-spatial datasets used in the main text and how we processed them for training with our limited computational budget.

## D.1 NOAA SST dataset

Huang et al. (2021) provide daily sea surface temperature data from September 1981 to the present day on a $1/4°$ global grid. We sub-sample this grid, converting it to a $4°$ resolution and also sub-sample the temporal horizon to get states every 10 days. We split the training and validation datasets using given years. 2000 to 2009 was used for training and 2011 to 2017 was used for validation. Notably, we deal with missing data over land by setting values to 0. The data was also normalized for training and validation.

## D.2 ClimSim dataset

The ClimSim dataset was originally developed by Yu et al. (2023) to provide training data for models predicting complex climate variables from more simplistic variables. However, we noticed it could also serve as an excellent candidate for exploring the learnability trade-off as data is reported every 20 minutes of the simulation. As such, we converted the low-resolution version the dataset, labeled `LEAP/ClimSim_low-res` for use in training our autoregressive models. We took a subset of the variables, namely the lowest $u$ and $v$ fields for prediction. The training and validation sets were taken from non-overlaping year-long periods. As with the SST data, we also normalised this dataset.

# E   Loss plots for Lorenz system

The Lorenz system is parameterized by three parameters $\theta_1, \theta_2, \theta_3$ corresponding to $\rho, \sigma, \beta$ in appendix C.1. In this section we expand the visualization from Fig. 6 to the other parameters for a given temporal horizon $T$. Note that in systems with more parameters such as the food web model, the plots become too complex.

We evaluate this function for $T \in [10, 500, 1000]$. See plots fig. 7, and fig. 8. In these figures we see, that the prediction already verified in a one dimensional loss in figure fig. 6 holds also for higher dimensional losses in dynamical systems.

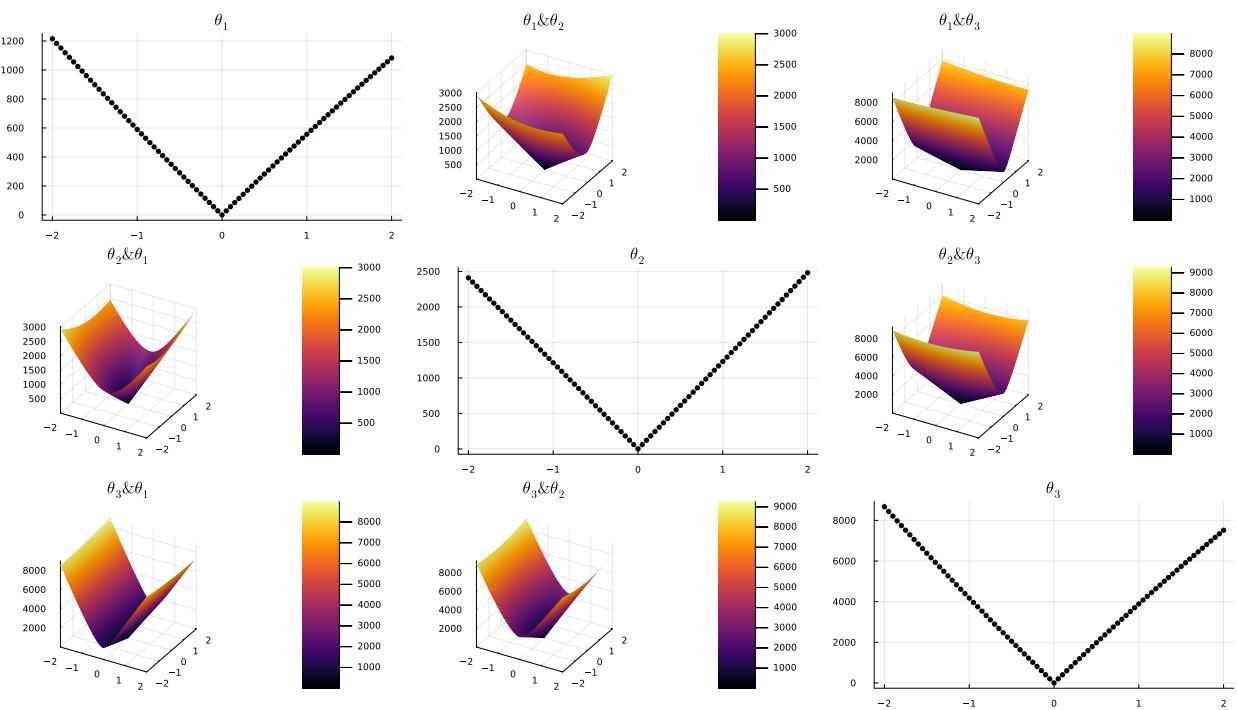

Figure 7: Surface plot of the piecewise loss function of the Lorenz system with $T = 10$

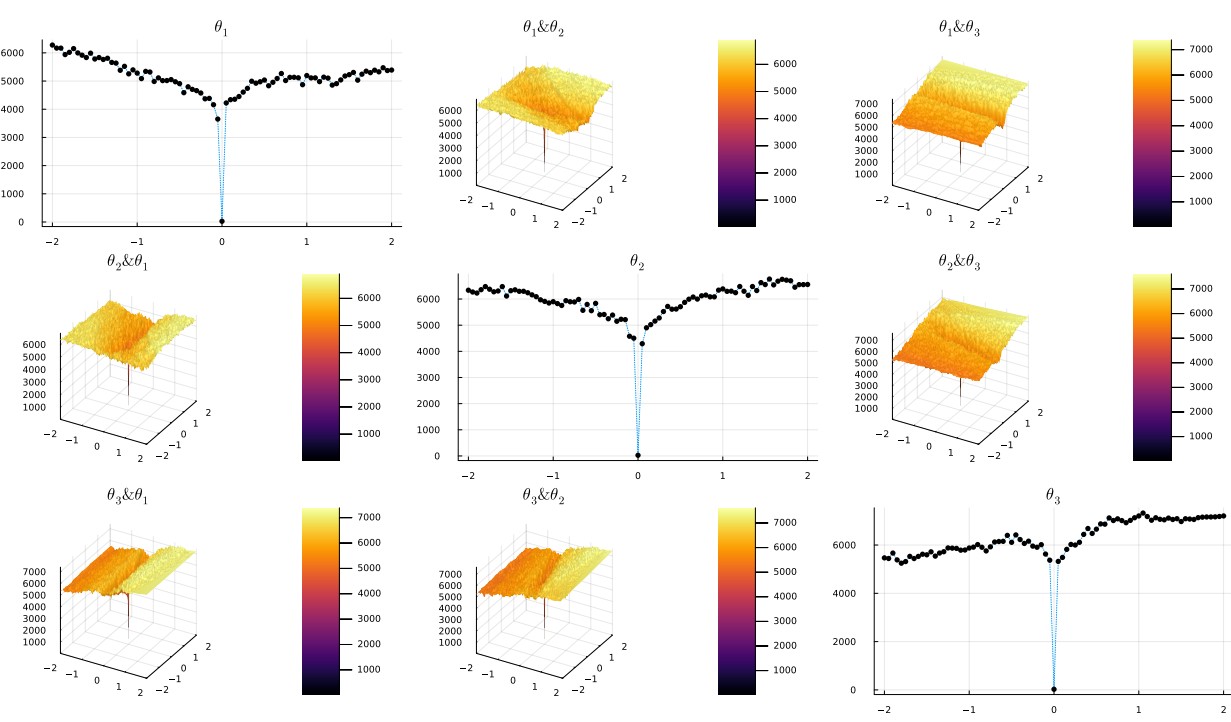

Figure 8: Surface plot of the piecewise loss function of the Lorenz system with $T = 1000$

## F   Training neural networks

In the following section we provide additional details on the training setup used for the four dynamical systems, as well as the networks used for the two climate-related datasets. We used a both MLPs andresidual MLPs with the same architectures.

### F.1   MLPs

We used two types of MLPs, a plain MLP without residual connections and one with residual connections.

**Definition F.1** (MLP)**.** We define a multi-layer to consist of an embedding layer ($W_{\text{embed}}$), an unembedding layer ($W_{\text{unembed}}$) and a series of blocks of the form:

$$\text{block}(x) = \text{ReLU}\left(W_{\text{block}}\text{LN}(x) + b_{\text{block}}\right) \tag{62}$$

where LN is a layer normalization. The network makes a given prediction by first applying the embedding layer, a series of blocks and then the unembed layer.

If we let $v$ be the input size of the example, we generally take $W_{\text{embed}} \in \mathbb{R}^{av \times v}$, $W_{\text{block}} \in \mathbb{R}^{av \times av}$ and $W_{\text{unembed}} \in \mathbb{R}^{av \times v}$ with varying values of $a$ and the number of blocks depending on the complexity of the dataset.

### F.2   Computer resources

We use less than 34000 core hours, or equivalent time of 1888 hours on a single V100 GPU.

### F.3   Hyperparameter choices

As a general rule, we tried to maintain some consistency between the hyperparameter choices for various datasets and architectures (for example we typically used a batch size of 512). However, due to the variability of both the data and the inductive biases, this was not always possible. For a detailed set of these, one would need to refer to the source code.

## G   Relationships between model complexity, learning rate, temporal horizon and noise on performance

In the autoregressive setting studied in this paper, there are many hyperparameters to set. One needs to decide, among other things, upon the complexity of the model, what learning rate to use and which temporal horizon to chose. To provide guidance on this front, we explored how performance was related to temporal horizon, learning rate and model complexity under different levels of observational noise. Specifically, we repeated the setup of section 5 for the ecology system with increasing label noise for a small, medium and large MLP using a total wall-time cutoff. We believe the following conclusions are evident from fig. 9:

- For (relatively) high forecasting horizons and learning rates the models fail to learn (red areas in all plots). This aligns with our theory showing that longer time horizons make the loss landscape harder to navigate, which can be alleviated with small learning rates.

- For any training horizon, the learning rates have an optimal value which is not at the extrema: high learning rates lead to a full failure to learn (red), but decreasing the learning rate can alleviate this. However, having a too low learning rate also decreases the performance of the model, likely because training has not been completed.

Those observations are both compatible with our theory and also fairly intuitive for machine learning practitioners. Taken together, we can conclude that with increasing time horizon and model complexity, the upper bound on effective learning rates decreases.

It is worth noticing that our theory was focused on deterministic systems with very small $\epsilon$ values, and both assumptions are not necessarily preserved in some subplots of Fig. 9. Thus, by comparing the different subplots we can reach the following conclusions:

- For small noise levels, the magnitude of the forecasting error for the optimal time horizon decreases with the model size (note that the error magnitudes decrease across rows). This can be understood as saying that the model can approximate the dynamics better ($\epsilon$ reduces).

- For high noise levels, all models achieve similar optimal performances, regardless of size. Our interpretation is that as $\epsilon$ is bounded from below by the randomness of the dynamics, the model size does not contribute to make better predictions

Taken together, those observations suggest that our theory is valid for small noise levels, but is not informative for large noise levels. However, at such levels no prediction is possible due to the inherently stochastic nature of the dynamics.

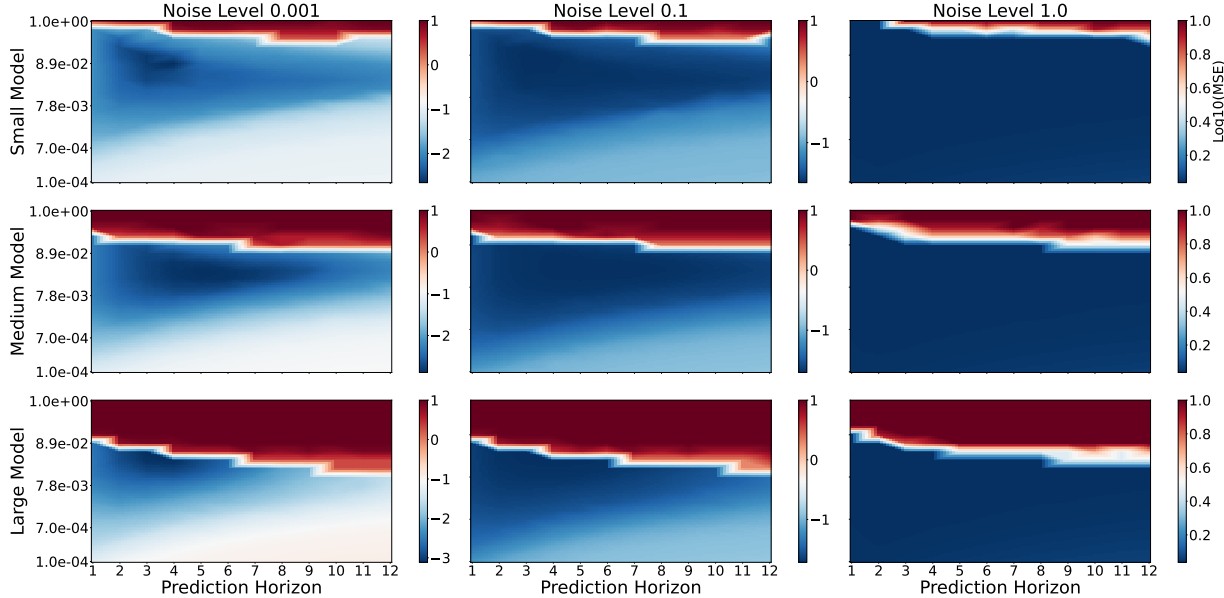

Figure 9: We trained our models at different levels of noise (columns) and with different parameter sizes (rows). For every combination, we tested different forecasting horizons (x-axis) and learning rates (y-axis). Note that the loss is normalized per plot.

## H  Increasing the temporal horizon increases performance of a sufficiently converged model

In this section, we provide empirical evidence for a down-stream effect that we should expect from theorem 4.6: For a sufficiently converged model, increasing the temporal horizon will increase performance, but a too large increase would make the gradients explode and disrupt the model. To do so, we consider MLPs trained on the ecology time series presented in the main text. We assume that we have a fixed training time $B$ that we split equally among different choices of temporal horizon. Given this split, we train the model by starting it on a prediction timestep $k = 1$, then switching to $k = 2$ and so on until we reach $k_{\max}$. This process is visualized in fig. 10.

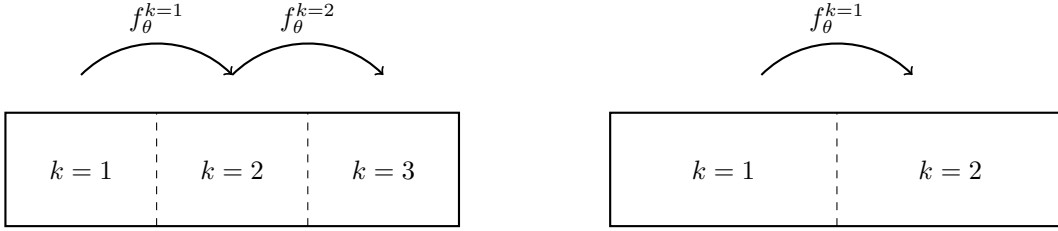

Figure 10: Experimental setup demonstrating the utility of increasing the temporal horizon of a converged model.

If our hypothesis is true, for sufficiently large values of $B$ up to a certain $k_{\max}$ we should get better performance from a greater splitting. fig. 11 reports the results of the experiment. As can be seen there, we do indeed get the expected trend, with greater total time favoring greater splitting with a larger total time horizon.

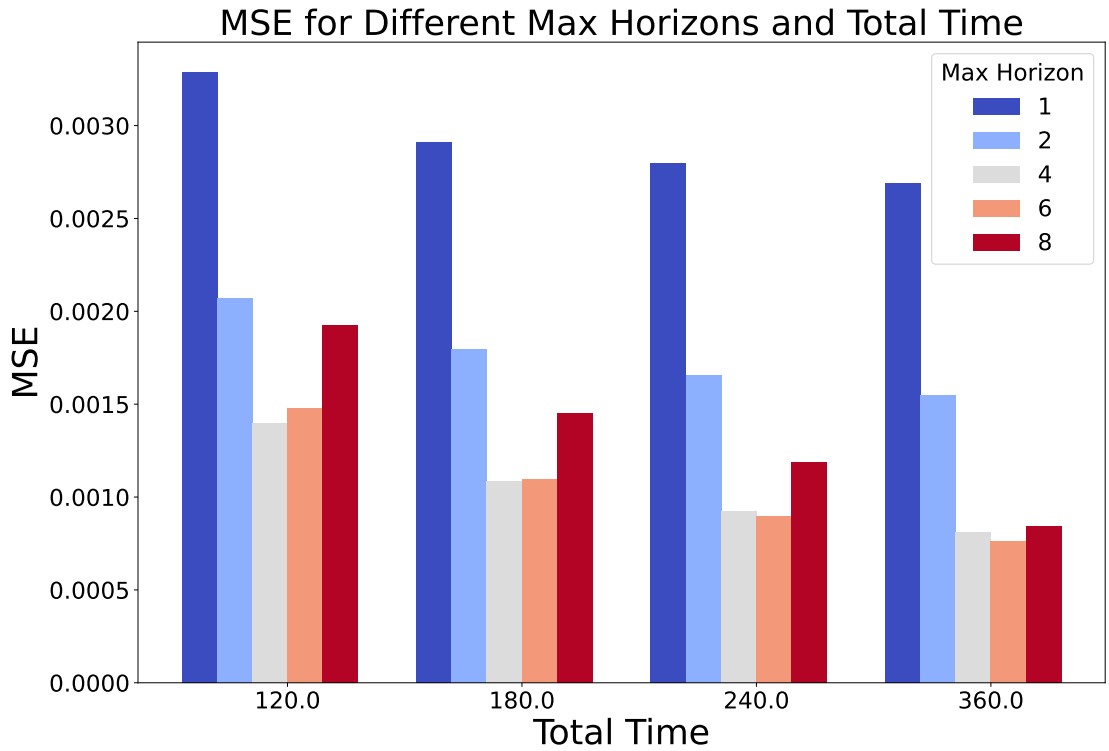

Figure 11: MSE compared to total time for various maximum horizons where training is split equally among time horizons up to the maximum horizon. We used the ecology system and computed the MSE

## I  Choosing time horizons and learning rates

Although we did not deem the results robust enough for inclusion in the main text, we did develop a scheme consistent with our theory for automatically deciding upon the training temporal horizon and learning rate. We would tentatively recommend this scheme to practitioners first applying eq. (3) to a well-behaved time series problem as given an initial $\eta_0$, it can sometimes out-performs the best $T$ but may also be able to uncover a reasonable choice for $T$ in cases where it does not perform as well.

### I.1  Motivation and specification of the algorithm

The scheme, detailed in algorithm 1, seeks to automatically choose $T$ (or equivalently $k$, the number of forecasting steps) and $\eta$ to overcome a common difficulty in gradient descent: the presence of plateaus. As we saw in theorem 4.7, the loss landscape becomes less flat with increasing $T$, suggesting that it should be possible to increase $T$ in order to deal with the plateaus. There is, however, some subtlety, because plateaus come in two forms: a minimum plateau and saddle-plateau. A minimum plateau is a region of low loss surrounded by higher losses, and a saddle-plateau is connected to some region of lower loss. In the later case, we'd like to speed up optimization at the expense of skipping small improvements that could be made in the region. In contrast, for a minimum plateau we would like to identify the best points within the plateau, and thus we would like to keep the size of the parameter update equal or smaller to explore a more informative minima. If we want to increase the step size to traverse a saddle-plateau, it suffices to increase $T$, and the gradient will increase either linearly or exponentially, depending on the underlying dynamics of the system as per proposition 4.3. Translating this logic into practice lead to development of the algorithm presented

---

**Algorithm 1** Iterative scheduling of $k$ and $\eta$

---

1: Initialize: $k \leftarrow 1$, $\eta \leftarrow \eta_0$, $\theta_{\text{prev}} \leftarrow \theta_0$, $\gamma \leftarrow 1.5 \cdot 10^{-4}$, look $\leftarrow$ True, $s = k_{max}/T_{max}$ succeeded $\leftarrow$ False
2: Set: $t_{start} \leftarrow$ time(), $t_{curr} \leftarrow$ time(),
3: **while** $t_{curr} - t_{start} < T_{max} * s$ **do**
4:     **if** look **then**
5:         $E \leftarrow 20$
6:         $\theta \leftarrow \theta_{\text{prev}}$
7:     **else**
8:         $E \leftarrow$ None
9:         time_limit $\leftarrow t_{max} - (t_{curr} - t_{start})$
10:     **end if**
11:     Train MLP with $T$ for $E$ epochs while recording $\mathcal{L}$, $\|\nabla_\theta \mathcal{L}\|$
12:     **if** validation loss improves **then**
13:         succeeded $\leftarrow$ True
14:         look $\leftarrow$ False
15:     **else**
16:         Adjust $\eta$ using exponential fit of trend in past $\|\nabla_\theta \mathcal{L}\|$
17:         $\theta \leftarrow \theta_{prev}$
18:     **end if**
19:     **if** early stop due to gradient using $\|\nabla_\theta \mathcal{L}\| < \gamma$ **then**
20:         $k \leftarrow k + 1$
21:         look $\leftarrow$ False
22:         succeeded $\leftarrow$ False
23:     **end if**
24:     $t_{curr} \leftarrow$ time()
25: **end while**

---

in algorithm 1. As can be read there, the algorithm works by waiting until we reach a flat region, looking ahead to see if we can obtain lower loss by increasing $T$ without decreasing $\eta$ and otherwise decreasing $\eta$ by fitting an exponential to the existing data (mimicking the scaling implied by proposition 4.3). Note that the algorithm is modified for the limit cycle system by replacing the exponential fit by a linear one.

## I.2 Results in comparison to normal training

In fig. 12, we compare algorithm 1 with networks trained using the same wall time with varying training temporal horizons and a constant learning rate on our four dynamical systems, changing the algorithm to fit a linear trend on the limit cycle. Note that some tuning to $\gamma$, $\eta_0$ and the wall time was needed to find examples where the iterative scheme showed promise. Nonetheless, as presented fig. 12, there are certain wall times, dynamical systems, learning rates and values of $\gamma$ one can choose which show that the iterative scheme can, in some cases, outperform or match the best choice of training temporal horizon without knowledge of that horizon beforehand. Some general observations from initial testing was that the scheme appeared to perform better in cases where $\eta_0$ was relatively low and $\gamma$ was also reasonably low. However, there was some non-robust sensitivity to $\gamma$ especially in the case of the limit cycle.

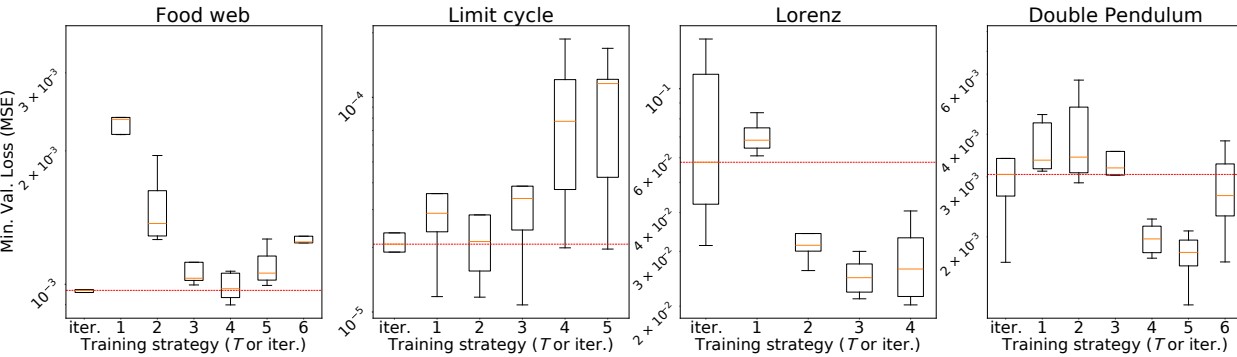

Figure 12: Comparison of iterative scheme to regular training on the four dynamical systems from the main text, as a box plot of the performances taken to be $\mathcal{L}(\theta, T)$ where $k$ is the maximum number of steps and the data is from the validation set. For each system and each time limit, we plot the temporal horizon for training (integers) or the iterative increase. The red line is the median of these scores for the iterative scheme. Note that $\eta_0$, wall time and $\gamma$ have been tuned in favor of the iterative scheme (although a full search was not applied).

