# OpenReview forum: "Temporal horizons in forecasting: a performance-learnability trade-off"
_TMLR — Accepted by TMLR_

### Review · Reviewer_EERN · 2025-07-12

**Summary Of Contributions:**

This paper studies how the training horizon affects forecasting performance in autoregressive neural networks (ARNNs). The paper shows that longer horizons improve generalization but make optimization harder, due to increased loss landscape roughness. The authors support this with theorical evidence and empirical experiments on synthetic dynamical systems. They conclude that the optimal training horizon depends on system dynamics and should be selected accordingly.

**Audience:**

Yes

**Broader Impact Concerns:**

No significant broader impact concerns.

**Claims And Evidence:**

Yes

**Requested Changes:**

Please address the weaknesses mentioned above.

**Strengths And Weaknesses:**

Strengths
* The paper addresses an important problem with clear motivation.
* The paper combines theoretical analysis with empirical validation in a well-balanced manner.
* The paper is well-written with a clear and logical structure across sections.

Weaknesses
* The theoretical analysis relies on strong assumptions such as ergodicity, stationarity, full observability, and complete state space coverage. These assumptions are rarely satisfied in real-world forecasting tasks, limiting the practical applicability of the paper contributions.
* Experiments are primarily conducted on synthetic dynamical systems. The absence of evaluations on widely-used benchmarks (e.g., Electricity, Traffic, Weather, and ETT time series [1]) reduces the generalizability of the conclusions and connectivity with existing works.
* While the authors present interesting findings based on ARNNs, it is important to note that ARNNs are not commonly regarded as a mainstream architecture in time-series forecasting, especially compared to RNNs or Transformers. This raises concerns about the general impact of the contribution.
To strengthen the significance of the proposed findings, it would be valuable to either (1) demonstrate that similar trends hold across other widely used architectures, or (2) provide more concrete evidence that ARNNs offer clear advantages over existing models in the context studied. This would clarify whether the observed effects are intrinsic to ARNNs or part of a broader phenomenon, thereby enhancing the relevance and impact of the paper.
* Horizon selection is still heuristic. Despite theoretical insights into the training horizon $T$, the practical selection still relies on grid search. The adaptive algorithm in Appendix H shows limited success, especially in chaotic systems, and does not fully address this limitation.
* It is difficult to find detailed descriptions of the values used in the experiments (e.g., the exact values of Tl and Th are unclear).
* In Figure 2, the y-axis label $r(T_h - T_l)$ is ambiguous; it is unclear whether this denotes a specific metric or is simply a typo.
* Figure 3 does not provide a clear explanation of z(T) in the caption. This forces readers to search the main text repeatedly to understand the notation, which is inconvenient.
* In Figure 4, the y-axis label uses undefined abbreviations in the main text, potentially causing confusion.
* Overall, the figures lack visual consistency (e.g., the placement of plot titles is not uniform across the figures).

[1] Autoformer: Decomposition Transformers with Auto-Correlation for Long-Term Series Forecasting, NeurIPS'21

---

> ### Author Response · Authors · 2025-08-02
>
> We thank the reviewer for their effort and feedback. We believe that the reviewer addressed important points that have improved our paper.
>
> We would like to highlight that we used simple problems and architectures because in those settings we can perform our analysis. Applying the same analysis to more difficult datasets and architectures or with fewer assumptions would likely require much more computer power and mathematical machinery. In turn, this would make our analysis not only harder to perform, but also much more technically complex, and thus difficult to understand.
>
> *The theoretical analysis relies on strong assumptions [...]*
>
> We would like to point out our assumptions are either unavoidable for almost any statistical learning theory, or can be relaxed (see below). However, just because something cannot be proven does not mean that it doesn’t hold, so we evaluated whether the U-shaped curve appears when those assumptions are broken. We took a small financial time series (financial data is generally believed to be stochastic, non-ergodic, and non-stationary), and still found a similar U-shaped curve.
>
> List of assumptions:
> - **Complete state space coverage:** This was a mistake on our part. We should have written that the data provides compehensive coverage of the state space (not complete). This cannot be avoided though, otherwise we are out of distribution.
> - **Stationarity:** Generally speaking, if the dynamics of the system change, we cannot predict them.
> - **Ergodicity:** This almost a mix the previous two. If we observe the whole (finite) state, and the system is stationary, the system is usually be ergodic. Replacing this makes proofs very complex, and often requires assumptions that are very subtle or technical (ex: Adams, arxiv:2408.01868)
> - **Full observability:** This assumption is not strictly speaking necessary, and we can relax it if we have a context window or recurrent connections (as hinted in the original text).
>
> We will extend the theory to drop the last assumption and add stochasticity.
>
> *Experiments are primarily conducted on synthetic dynamical systems [...]*
>
> We would like to extend our experiments to other problems, but this is computationally very hard. Even in the simply dynamical systems with MLPs, we find that we already encounter limitations in visualizing Thm. 4.4 and 4.5, as we noted in the captions of Fig. 2, 3.
>
> To make our computational analysis we need good forecasts (small epsilon in the theorems), together with a parameter space that is small enough to be studied in practice. This is unlikely to happen in existing benchmarks without extensive compute resources and more complex architectures (with more complex visualizations of Thms. 4.4/4.5)
>
> *While the authors present interesting findings based on ARNNs [...]*
>
> We would like to thank the reviewer for this comment, as it made us improve our paper.
> We do think that ARNNs are SOTA in some tasks (Graphcast, by Lam et al., Science 2023, Kaufmann et al., Nature 2023).
> Also, we note that our results hold in mechanistic models (non neural networks), suggesting that this is very general (sec. 5.2 and Appendix C), and the U-shaped curve appears in related literature (see related works, and the response of Reviewer 4w7z).
>
> However, we do agree that it would make our results more impactful if we extend our theory to other architectures.
> We realized that we can extend the definition of epsilon-bounded to include RNNs and Transformers through delay embedding theorems, and in the process drop the full observability assumption. We will add those to the manuscript.
>
> *Horizon selection is still heuristic. [...]*
>
> We thank the reviewer for pointing this out. In short, the comparison of our algorithm with the grid search is not fair to our algorithm (see explanation below). We did not report this properly, and we apologize for the omission.
>
> **Comparison:** In Appendix H we gave all horizons the same computing time as our algorithm. However, how would the algorithm know that it had to use T>1 to begin with? With a grid search, the algorithm would spent more resources to discard the earlier time horizons (specifically, T+1 times more resources). It is also not possible to run an initial search with limited resources: As the optimal temporal horizon depends on the computational resources used (App. G), a resource-limited search would not be informative.
>
> The advantage of our algorithm is that it re-uses the training of low $T$ to initialize the model at higher $T$. Although it cannot compete with an algorithm that has the *a priori* information about the optimal $T$ (in chaotic systems), it does better than the simplest approach.
>
> Thus, we believe that developing strategies for finding optimal T is a paper on its own. If that paper is written, our theory could be a valuable guide.
>
> *Clarity and labeling comments.*
>
> We appreciate the reviewer’s detailed review, and we will add the explanations and correct the figures.

---

> > ### Comment · Reviewer_EERN · 2025-08-23
> >
> > I thank the authors for their rebuttal. I recognize the contribution of this work in clarifying how the training horizon affects both generalization and optimization, and I consider the theoretical and empirical analysis to be valuable.
> > While I still think that stronger connections to other time-series forecasting studies and widely used benchmarks would further enhance the impact, this does not diminish the overall contribution. I therefore lean toward recommending acceptance.

---

### Review · Reviewer_GSFB · 2025-07-27

**Summary Of Contributions:**

This paper investigates the trade-off between forecast horizon length and learnability in recurrent neural networks (RNNs, though the paper seems to make a new definition and call them auto-regressive neural networks...) trained on dynamical systems. The authors argue that longer training horizons provide better generalization, especially in chaotic systems, but make optimization harder due to a rougher loss landscape. They offer theoretical analyses of gradient and Hessian scaling with respect to the forecast horizon, and validate their results empirically on synthetic and real-world datasets.

**Audience:**

Yes

**Broader Impact Concerns:**

NA.

**Claims And Evidence:**

Yes

**Requested Changes:**

I do not believe the current version is publication level and requires substantial revisions to meet either of the two criteria. Here are my suggestions for a revised submission:

1) Correct the definition of a dynamical system. The current formulation inverts time and misrepresents standard terminology. Clearly distinguish between continuous and discrete-time systems and define them properly.

2) Revise the taxonomy of dynamical systems. The four-category classification (“stable, unstable, limit cycles, chaotic”) is incorrect and conflates local stability with global system types. I would avoid simplification of this form, which was quite unpleasant for someone with a physics background to read.

3) Clarify model architecture terminology. The term “ARNN” is used in a non-standard way. Clearly differentiate between feedforward autoregressive models, RNNs, and Transformers, and avoid suggesting that only the former are autoregressive. The definition in Eq. (2) IS an RNN.


4) Acknowledge that exponential forecast divergence occurs even with perfect models and address the role of noise more directly. The exponential gradient and loss growth described in the main theorems are not specific to learned models, they arise directly from chaotic dynamics and would be observed even with true parameters. A similar argument can be made for the limit cycle. This weakens the claim that long-horizon training yields fundamentally better generalization. Given that exponential divergence arises even in perfect models, the paper’s main results appear to restate known properties of chaotic dynamics rather than introduce new insights into training RNNs. Many real-world systems are stochastic. The theory should either explicitly incorporate noise into the core results (not just appendices) or clearly state that the current framework applies only to noise-free systems, in which case I am not sure if it satisfies the interest to the general TMLR audience criteria.

5) Rewrite the manuscript for clarity. The core results should be presented in the main text, not appendices. The main text is very hard to read and digest in its current form without looking at the appendices regularly. The work has more space to work with, and potentially can be longer than 12 pages as TMLR allows it.

6) Few data points is not enough to claim exponential/linear divergence: Plots include only a few data points and no rigorous evidence that the stated scaling takes place. The scaling claims (exponential vs. linear) cannot be substantiated with only 5–6 data points and should be supported by formal curve fitting or statistical analysis.

7) Ignoring advantages of single-step paradigms: Several works have surfaced recently that use single-step paradigms to train RNNs with convex optimization within seconds, what would have taken days with BPTT. What the theoretical results are saying is that accounting for longer time windows does not prevent any further correctness than just training with a single-step prediction errors if we have a noisy model (which is the case in practice). So, the argument made by the paper for me is actually the opposite of what is claimed.

**Strengths And Weaknesses:**

## Strengths

This is an interesting topic in general, and quite timely with the recent successes of state-space models like Mamba.

## Weaknesses

The writing has several issues with unsubstantiated or outright incorrect claims (see below). The experiments involve only a few data points, and there is no consideration of the noise which may wash out the main results reported in this paper. For instance, noisy limit cycles or chaotic networks will likely have the same scaling of error in time, regardless of how far ahead they are trained to predict.

---

> ### Author Response · Authors · 2025-08-01
> **We will clarify the paper, but believe that the review does not cover the full paper.**
>
> We appreciate the critical review, and agree that the reviewer raises some good points about clarity. However, we believe that there are some incorrect or confusing points in the review. Here the main ones:
> - The reviewer argues that training with single step predictions can be better (point 9), and that our claim for improved generalization for longer horizons is weakened (point 4). Yet, the improved generalization is an empirical (not theoretical) fact that we show in Fig. 4,5, and we discuss it in the related works section.
> - The reviewer states in their summary that our results are about gradients and hessians exclusively. Yet, our main theoretical results (Thm 4.4, 4.5) are about loss landscape geometry (ex: number of local minima).
> - The reviewer states that the noise is only discussed in the Appendix, but there are two paragraphs on section 5 of the main text about it.
>
> We will address the points raised by the reviewer.
>
> *Summary*
>
> The summary is missing the results about loss landscape geometry (Thms 4.4 and 4.5), and does not acknowledge that we discuss the role of noise in the main text (Sec. 5).
>
> *Points 1,2*
>
> We will update the definitions of dynamical systems, and clarify that we refer to attractors of dynamical systems, not the systems themselves.
>
> *Point 3*
>
> We apologize for the lack of clarity. We picked a simple architecture to make our theoretical study, but we do agree that it could lead to misunderstandings. Specifically, we are referring to systems that can be autoregressive during training. In this context, transformers are usually not *trained* in an autoregressive manner, as they make a single-step prediction during training (two notable exceptions are discussed in Related works).
>
> Also, Eq. 2 is only an RNN if T>1. This is important, because a model can be trained with T>1, but used during inference with T=1 (see Sec. 5). If Eq. 2 was the definition of RNNs, there would be models that are RNNs during training but not during inference (which in our opinion is not how RNNs are usually defined).
>
> We will specify that we refer to models that can be autoregressive during training (but do not need to be during inference), and change the notation to Autorregressively Trained Neural Networks (ARTNN). We will also specify that by RNNs we mean networks with feedback loops in the hidden state.
>
> *Point 4, acknowledgements*
>
> We explicitly explained the exponential divergence in the definition of chaotic dynamical system. It is strange to claim that we did not acknowledge it.
>
>
> *Point 4, exponential divergence*
>
> Our two main theorems (4.4 and 4.5) relate to loss landscape geometry. In our opinion, those are not meaningful in perfect models. In fact, we explicitly stated in the main text (sec. 5.2) that Thm. 4.4 does not apply to perfect models. Thus, we disagree with the assessment that the results simply restate properties of dynamical systems. We can downgrade our first theorem (Thm. 4.2) to a proposition.
>
> We note that longer training horizons do yield better generalization, as we show explicitly in Fig. 4,5. This also seems to appear in existing literature, (as we discuss in related works). We are unsure as to why the reviewer argues that this claim is weakened.
>
> *Point 4 (noise) and Point 5*
>
> We agree that the effects of noise are important in real world systems, but note that we discussed this in the main text (page 9, last paragraphs, Sec. 5), not just appendices. We can shift that discussion to the theory section.
>
> *Point 6*
> The goal of Fig. 1,2,3 is to illustrate that the theorems affect the loss landscape rather than an empirical proof (as noted in Sec. 3.2). Specifically, we picked timesteps that are close to the range in which we show the U-Shaped curve that appears in Fig. 5 (which is the main problem we address).
>
> For Fig 1, we could indeed try to fit a curve, but this Figure relates to known mathematical properties of dynamical systems (as the reviewer noted), so it does not seem necessary to prove them empirically. For Fig. 2,3, we cannot see those trends in any case due to experimental limitations (as noted in the figure captions).
>
> *Point 7*
>
> We are unsure as to whether the works referred by the reviewer apply to the models that we discuss, and note that some SOTA models are trained in an autoregressive manner (ex. GraphCast). Furthermore, our theory (4.4 and 4.5) includes properties of the loss landscape, which should not depend on the learning algorithm. However, we would gladly add those references, and we kindly ask the reviewer to provide them.
>
> *Point 7, one-step in noisy systems*
>
> We agree that the optimal time horizon would likely be T=1 with a very noisy model (as shown in Appendix F), but disagree with the statement that all practical systems are noisy to that level. This is shown in Fig. 4, and in the related works. To address this point further, we tried a small financial time series (which are often considered to be very noisy) and found that T>1.

---

> ### Comment · Reviewer_GSFB · 2025-08-02
>
> Thank you for your kind reply. I will provide a more comprehensive explanation on monday. For now, I would like to emphasize two key aspects:
>
> 1) Could you please update the manuscript so that we can assess the updated version, as opposed to committing to future updates?
>
> 2) My main concern is that the main contributions (exp and linear scaling) as stated and emphasized in abstract and throughout the paper feels trivial. My argument was that if you were to start with a noisy model and infer, the errors accumilating over time would follow exponential (or linear) scaling to begin with. So it is not surprising you need to train that much longer to account for it. It is also likely the case that this rapid buildup of the noise limits the maximum trainable time steps. Maybe stating this explicitly would also help.
>
> Regardless, I would rephrase the contributions in a way (one emphasizing many interesting aspects raised in these responses in the abstract) that they are interesting to the broader TMLR audience and doesnt come off as trivial textbook results (which they are not, I agree!). For instance, having a chaotic model trained on short time-scales making exponentially worse errors for long time scales is not as interesting (which is expected, since the same model with noise has also exponentially increasing distances between two trajectories with equal starting points) as the finding that noise and loss landscape set a practical cap for how long ahead one can train. I think the latter is very interesting and could be more emphasized.
>
> I hope this is helpful

---

> > ### Author Response · Authors · 2025-08-04
> >
> > We appreciate the positive review and agree that our initial submission did not emphasize the more insightful contributions. We did find this helpful, thanks!
> > And this time we did update the manuscript

---

> > > ### Comment · Reviewer_GSFB · 2025-08-06
> > >
> > > I admit that the new abstract and the writing on the background are much improved. I personally could make further suggestions to improve the manuscript, but at this point, given that how other reviewers like this work, I will update my position to acceptance. Here are more details about the points I raised and how they changed after author responses:
> > >
> > > 1-2) I would likely still refrain from defining a taxonomy here. For instance, the sentence "thus attractors forming manifolds, which can be categorized in limit cycles and chaotic attractor" is missing the slow-point manifolds. I just dont think this taxonomy is needed here, you can simply state you believe (and many including myself would agree) that these two particular cases are interesting and are the topic of this study.
> > >
> > > 3) I still do not get this distinction. There are certainly RNNs trained with single-step predictions and are used to infer single time step ahead. I will refer to this particular review: https://www.nature.com/articles/s41583-023-00740-7. For instance, this work includes both single-step ahead predictions and training: https://proceedings.neurips.cc/paper_files/paper/2022/hash/9877d915a4b4f00e85e7b4cfdf41e450-Abstract-Conference.html
> > >
> > > 4) I think my comment was not clear here. But, this is basically the point I raised above in the follow-up with point 2).
> > >
> > > 5) I would have liked to see a clear hypothesis about noise being tested within the results section, whereas the manuscript discusses it as part of an appendix and as a paragraph in discussion. I believe the trade-offs caused by noise can be a bit more clearly illustrated. For instance, I don't really understand (well, I do, but it takes time and effort) the take-away message from Figure 9. That being said, I will once again not push this direction further, as authors stated elsewhere that this direction is considered future work. I think the work is still interesting without it, though it is a bit of a bummer.
> > >
> > > 6) Can you maybe cite the appropriate work whenever you make claims such as "the food web is chaotic, but only in very long timescales, but for short timescales it is effectively a limit cycle?" This would be easier and cleaner than the reader searching for experimental evidence.
> > >
> > > 7) Well, I am a neuroscientist and I am thinking of neuroscience work that is under the umbrella term of "data-constrained RNNs." People use all sorts of algorithms for their training, and the goal is often not to be SOTA (in fact, it is not even well-defined what it means to be SOTA), rather to fit interpretable models that can be reverse-engineered and their biological hypotheses studied. You can refer to the box 3 and works therein in here: https://www.nature.com/articles/s41593-025-02031-z.
> > >
> > > In my personal taste, I would have likely done a few revisions. Within my field, dynamical system theory is regularly studied as part of the studies of neural computation, and there are still some phrases that many would likely object to as they are reading this work. Alas, it is not my decision, and the authors did not write this paper for people in my field who may or may not particularly like some of the blanket statements. Hence, I will no longer push forward for additional clarity. Good luck!

---

> ### Author Response · Authors · 2025-08-08
>
> We thank the reviewer for their assessment and further suggestions. We agree that we can simplify our text.
>
> We address their points one by one.
>
> 1-2) We did avoid the taxonomy in the current version. Thanks for the suggestion!
>
> 3) Since we extended our theory to (what we understand as) RNNs and transformers, we decided to try to avoid further confusion and refer generally to AR models, and to RNNs and Transformers as specific architectures that can be AR. Our theory is centered on feedforward AR networks, but it should generalize to the RNNs/Transformers (and the first step to do so is provided). We do not fully develop the theory or experiments, but we believe that with Prop. A.20 it is a straightforward process (albeit long and with many definitions and assumptions)
>
> 5) We appreciate the reviewer's comprehension.
>
> 6) Well spotted. We did it by visual inspection, but agree that this was not rigurous. As it is also unnecessary, we deleted those statements altogether.
>
> 7) We appreciate the clarification. It is possible that the goal in neuroscience might be part of the source of our confusion. If we define AR models as those that use their own predictions or outputs, and RNNs networks with feedback loops in the hidden state, then the question is whether the hidden state counts as prediction/output.
> In our understanding, the hidden state in RNNs does not constitute a prediction/output of the model, (it's hidden), thus an RNN would only be autoregressive if if was iteratively applied on its own outputs (not hidden states).
> In neuroscience the (hidden) state of the neurons is precisely what is interesting about the model, and thus it would make sense to consider it an output/ prediction.
> In any case, we believe that the current version avoids the conflict, or at least makes it clear that the results are similar even if the definitions were different.
>
> A note on neuroscience that might be interesting for the reviewer. We believe that our results could have some relevance to another line of neuroscience literature. Specifically, replays of trajectories (usually studied in hippocampus, but present in cortex too), because some studies show that prolonging them (temporally) leads to better learning in simple tracks (Fernández-Ruiz, et al. "Long-duration hippocampal sharp wave ripples improve memory." Science). Anyway, this is definitely outside of our paper's scope, just a note.

---

> ### Comment · Reviewer_GSFB · 2025-08-08
>
> I appreciate the update. I will give a detailed read to the final version, hopefully not as a reviewer but as a reader of TMLR. At least, I personally recommend acceptance now. I thank the authors for the changes, it seems from a quick look that the work has improved significantly.

---

### Review · Reviewer_4w7z · 2025-07-27

**Summary Of Contributions:**

Sorry for writing my review late.

This work theoretically studies the implication for training when we train a model to predict a very long horizon.

**Audience:**

Yes

**Claims And Evidence:**

Yes

**Requested Changes:**

Nothing really

**Strengths And Weaknesses:**

Strength:
(1) The paper is very well written and easy to understand

(2) The theory, while simple, is very insightful, and directly answers the question of why it is often difficult to train a model to predict a far away horizon

(3) This perspective is also novel, and I believe it could be extended to analyze other problems (such as LLM training)

(4) The theory is quite comprehensive, covering the growth of gradient, Hessian and number of local minima, basically everything one should care about in learning these dynamical systems

(5) A key message of the paper is that the longer the prediction horizon is, the more difficult it is to train the model -- a well observed (and almost universal phenomenon). Therefore, I believe that this is simple theory that partially tackles one of the most important problems in this field

Weakness:
Nothing major. The figures are well plotted, the paper is well explain. The theory is clean and easy to understand

Interesting and great work!

---

> ### Author Response · Authors · 2025-08-02
>
> We thank the reviewer for their kind assessment!

---

### Decision · Action_Editor_uRbn · 2025-09-06

**Recommendation:** Accept with minor revision

**Additional Comments:**

The initial reviews of the paper were mixed. On the positive side, the reviewers acknowledged the papers clear motivation and comprehensive theoretical analysis. However, they also raised a number of important concerns, such as the papers strong assumptions, simple architectures, and limited evaluation. In their rebuttal the authors addressed many of these points, clarifying the need for their mathematical assumptions / experimental setup and positioning their theoretical insights relative to more modern network architectures. While the practical impact of the presented framework is modest, this is often the nature of theoretical works and the visible effort to push these inherent limitations as far as possible - even if at times into murky territory - must be appreciated. After the rebuttal phase all three reviewers recommend acceptance. I share this general sentiment but want to emphasize that the presentation does not yet meet the requirements for acceptance. This includes:

- Structural issues: please follow standard practice and separate theory and experiments. Please introduce a clearly titled experiments section, which should include Figures 1-3 plus accompanying descriptions and the relevant content in section 5.
- Contribution issues: it is at times difficult to distinguish between prerequisites/background and original contributions. For example, is the concept of an $\epsilon$-bounded region a novel technique introduced in this paper? Please ensure that all pre-existing theory is clearly identified and be explicit if something is novel.
- Self-containedness issues: there are instances of non-trivial mathematical concepts (e.g., “locally unstable trajectories”) without proper definition.
- Orthography/grammar/notation issues: there are numerous preventable typos, grammatical errors, and notation/index inconsistencies.

I urge the authors to revise their manuscript according to the points above, which are not optional and will be taken seriously. After these changes have been made, the manuscript will be recommended for publication.

**Audience:**

Yes

**Audience Explanation:**

The presented framework is mainly of theoretical value and its practical impact will be limited. Especially the paper’s strong assumptions and simple architectures are a bottleneck in this regard. However, it can serve as a starting point for future research that might close these gaps. The paper already lays the groundwork for such extensions, including preliminary work on stochastic systems, modern backbones (e.g., transformers), and horizon-selection algorithms. It will therefore be of interest to readers wanting to build upon this work, and there are several options to do so.

**Claims And Evidence:**

Yes

**Claims Explanation:**

The paper is grounded in an extensive theoretical framework. The appendices, in particular, provide significant context for the results in the main paper and explore new paths along which the presented analyses can be extended. The theoretical contributions are validated in a series of synthetic scenarios with known properties. Additional experiments on real-world data show that some theoretical insights carry over to more realistic environments.

---

> ### Author Response · Authors · 2025-09-26
> **New version updated.**
>
> Dear Editor,
>
> We appreciate the editor's thorough revision and the overall positive evaluation. We have addressed the comments raised in a new version. More specifically, we have:
> - corrected typos and grammar mistakes
> - Explained any non-trivial mathematical concepts, or deleted references to them when they were not relevant.
> - Clarified which contributions are new and which are already known. For example, the $\epsilon$-bounded regions are a new definition that allows us to apply relatively standard concepts in geometry.
>
> However, we believe that the existing structure is better suited to convey the contributions of the paper, so we have kept it as is for now. Since Fig. 1 is a visualization of Prop.1, Fig. 2 is a visualization of Thm. 1, and Fig. 3 is a visualization of Thm. 2, it makes sense to put these together.
> If we use the classical theory/experiment structure, the readers would have to first go through the theory without visualization, then read the experiments while going back to the theory to contrast the formula with the plots. We tried this in an earlier version and found out that it gave a worse reading experience (including to some colleagues who gave feedback). We also want to point out that the reviewers found the paper easy to read and well structured.
>
> That being said, if the editor decides that this is non-negotiable we would change the structure.